# Quantitative prediction of variant effects on alternative splicing in *MAPT* using endogenous pre-messenger RNA structure probing

Jayashree Kumar[1,2], Lela Lackey[1,3], Justin M Waldern[1], Abhishek Dey[1], Anthony M Mustoe[4], Kevin M Weeks[5], David H Mathews[6], Alain Laederach[1,2]*

[1]Department of Biology, University of North Carolina at Chapel Hill, Chapel Hill, United States; [2]Curriculum in Bioinformatics and Computational Biology, University of North Carolina at Chapel Hill, Chapel Hill, United States; [3]Department of Genetics and Biochemistry, Center for Human Genetics, Clemson University, Greenwood, United States; [4]Verna and Marrs McClean Department of Biochemistry and Molecular Biology, Therapeutic Innovation Center (THINC), and Department of Molecular and Human Genetics, Baylor College of Medicine, Houston, United States; [5]Department of Chemistry, University of North Carolina at Chapel Hill, Chapel Hill, United States; [6]Department of Biochemistry & Biophysics and Center for RNA Biology, School of Medicine and Dentistry, University of Rochester, Rochester, United States

*For correspondence:
alain@unc.edu

**Abstract** Splicing is highly regulated and is modulated by numerous factors. Quantitative predictions for how a mutation will affect precursor mRNA (pre-mRNA) structure and downstream function are particularly challenging. Here, we use a novel chemical probing strategy to visualize endogenous precursor and mature *MAPT* mRNA structures in cells. We used these data to estimate Boltzmann suboptimal structural ensembles, which were then analyzed to predict consequences of mutations on pre-mRNA structure. Further analysis of recent cryo-EM structures of the spliceosome at different stages of the splicing cycle revealed that the footprint of the B[act] complex with pre-mRNA best predicted alternative splicing outcomes for exon 10 inclusion of the alternatively spliced *MAPT* gene, achieving 74% accuracy. We further developed a β-regression weighting framework that incorporates splice site strength, RNA structure, and exonic/intronic splicing regulatory elements capable of predicting, with 90% accuracy, the effects of 47 known and 6 newly discovered mutations on inclusion of exon 10 of *MAPT*. This combined experimental and computational framework represents a path forward for accurate prediction of splicing-related disease-causing variants.

## Editor's evaluation

This manuscript will be of interest to biologists who study RNA structure-function relationships in a broad range of systems, splicing researchers, and RNA structure bioinformaticians. An integrative analysis of RNA structure probing, model-based RNA folding energetics, cryo-EM data, and protein binding sequence motifs serves as the basis for a comprehensive, accurate, and robust framework for predictive models of splicing dynamics in a well-studied system. The modeling is leveraged by in silico mutagenesis that reveals novel insights into the mechanisms and tradeoffs that underlie the impact of disease-associated mutations on alternative splicing.

## Introduction

Precursor mRNA (pre-mRNA) splicing is a highly regulated process in eukaryotic cells (*Wang and Burge, 2008*). Numerous factors control splicing including *trans*-acting RNA-binding proteins (RBPs), components of the spliceosome, and the pre-mRNA itself. Pre-mRNA structure is a key attribute that directs splicing, particularly alternative splicing, but we have a limited understanding of pre-mRNA structure-mediated splicing mechanisms (*Taylor and Sobczak, 2020*). It has proven challenging to develop quantitative models capable of predicting splicing outcome, specifically the percent spliced in (PSI) for alternatively spliced exons. It is especially difficult to predict outcome alterations due to genetic variation at exon-intron junctions because mutations affect both the binding by RBPs and also pre-mRNA structure (*Tazi et al., 2009*).

The consequences of mutations on pre-mRNA structure are difficult to predict. First, little is known about native pre-mRNA structure because pre-mRNAs are relatively short-lived in cells (*Herzel et al., 2017*). Only recently has high-resolution in-cell experimental characterization been applied to pre-mRNA structure determination (*Mustoe et al., 2018a*; *Sun et al., 2019*; *Liu et al., 2021*; *Bubenik et al., 2020*). Second, it is not clear which structures within a pre-mRNA modulate spliceosome assembly and activity. Finally, quantitative measures for the relative weighting of RBP affinity for individual motifs within a pre-mRNA relative to the importance of pre-mRNA structure are lacking.

In this study, we exploited several technical developments that address these issues to develop an integrated, RNA structure-based framework that accurately predicts splicing outcomes. We measured endogenous pre-mRNA structure in cells taking advantage of recent developments in mutational profiling (MaP) approaches for read-out of chemical probing data (*Homan et al., 2014*) with targeted amplification of specific exon-intron junctions. This novel approach enabled us to obtain single-nucleotide RNA (snRNA) structure probing data for endogenous pre- and mature mRNAs in the same cell. Our RNA structure modeling considers the equilibrium between multiple alternative structures (*Dethoff et al., 2012*; *Lai et al., 2018*) and employed data-guided Boltzmann suboptimal sampling (*Spasic et al., 2018*) to predict free energies of unfolding for structures in the ensemble. We additionally leveraged recent high-resolution structures of the spliceosome at various stages of the splicing cycle to deduce the effective spliceosomal footprint on pre-mRNA (*Zhang et al., 2019*), quantitative analysis of exonic and intronic splicing enhancers (ESEs and ISEs)/silencers (ESSs and ISSs) (*Fairbrother et al., 2002*; *Wang et al., 2004a*; *Wang et al., 2012*; *Wang et al., 2012*), and a β-regression weighting (*Ferrari and Cribari-Neto, 2004*).

For validation of our framework, we studied the effects of 47 experimentally measured mutations near the exon 10 – intron 10 junction of the human *MAPT* gene, which encodes the Tau protein (*Park et al., 2016*; *Silva and Haggarty, 2020*). Exons 9, 10, 11, and 12 encode the critical microtubule binding repeat domain in Tau. Exons 9, 11, and 12 are constitutively spliced, but exon 10 is alternatively spliced resulting in *MAPT* isoforms with either four microtubule binding repeats (4R) or three repeats (3R) when exon 10 is included or skipped, respectively. The normal ratio of 3R to 4R isoforms is approximately 1:1 (*Hefti et al., 2018*). Twenty-nine clinically validated disease-causing mutations have been identified in the region of the exon 10 – intron 10 junction (*Stenson et al., 2003*). These mutations result in impaired Tau function and are implicated in neurodegenerative disease (*Spillantini et al., 1998*; *Hutton et al., 1998*; *Clark et al., 1998*; *Rizzu et al., 1999*; *Goedert et al., 1999*). Although some mutations alter the Tau protein sequence (*Mirra et al., 1999*; *Iseki et al., 2001*), 20 of the disease-associated mutations deregulate *MAPT* pre-mRNA splicing altering the ratio of 3R to 4R (*Hutton et al., 1998*; *D'Souza et al., 1999*; *Hasegawa et al., 1999*; *Jiang et al., 2000*). The effect of an additional 27 mutations on exon 10 inclusion has been experimentally determined using cell-based splicing assays (*D'Souza and Schellenberg, 2000*; *Tan et al., 2019*; *Grover et al., 1999*). The exon 10 junction is the best experimentally characterized junction of clinical importance in the human genome and is thus an excellent system for developing forward-predictive models of splicing. Our work provides a framework for integrating endogenous pre-mRNA structure probing data with a structure-based understanding of spliceosome assembly and *trans*-acting RBPs to qualitatively predict the effect of mutations at exon-intron junctions on splicing.

## Results

### *MAPT* 3R and 4R mRNA isoforms are expressed at a consistent 1:1 ratio across tissues

To confirm that *MAPT* pre-mRNA splicing results in a 1:1 ratio of alternatively spliced isoforms (*Goedert et al., 1989*; *Andreadis, 2005*) in a large population, we analyzed RNA-sequencing data from the genotype-tissue expression (GTEx) database (*Lonsdale et al., 2013*). We analyzed data from tissue types with median *MAPT* transcripts per million (TPM) greater than 10 (*Figure 1—figure supplement 1A*) and calculated the PSI value for exon 10 for each sample (*Figure 1—source data 1*; Materials and methods). We examined data from 2315 tissue samples from 375 individuals of median age 61 (*Figure 1A* and *Figure 1—figure supplement 1B*). A PSI of 0 indicates that none of the *MAPT* transcripts in a sample included exon 10 (3R), whereas a PSI of 1 corresponds to exon 10 inclusion in all transcripts in a sample (4R).

PSI for exon 10 varied across tissue types and within and between individuals. However, 75% of samples were within an SD of the median PSI of 0.54, demonstrating that the 3R to 4R isoform ratio was close to 1:1 among individuals and across tissues. Within the brain, the pituitary gland demonstrated the largest variation in PSI and the cerebellum the least variation. The pituitary gland also had the lowest median PSI (0.38). However, the median PSI differed by no more than 0.25 across all brain tissues. Interestingly, although *MAPT* function in breast tissue is not understood compared with its function in the brain, there was greater variation in PSI in breast tissue, and the median PSI in breast tissue was lower than in the pituitary gland (*Figure 1—figure supplement 1B*). There was also a large amount of variation within tissues of an individual (*Figure 1—figure supplement 1C*), although there was significantly greater variation between than within individuals (see *Supplementary file 1* for ANOVA table). Furthermore, exon 10 inclusion variability (0.2) was between the variability for a *MAPT* constitutively spliced exon (0.1) and another *MAPT* alternatively spliced exon (0.3) (*Figure 1—figure supplement 2*). As levels of RBP expression varied considerably across individuals and tissues (*Figure 1—figure supplement 3*), sequence and structural features of the *MAPT* pre-mRNA likely regulate inclusion of exon 10.

### Structures of 3R and 4R *MAPT* mature mRNA isoforms are similar and mostly unstructured

The structures of the mature 3R and 4R isoforms and *MAPT* pre-mRNA have not been assessed in their endogenous context in cells. Here, we used dimethyl sulfate probing read out by mutational profiling (DMS-MaP) as described previously (*Mustoe et al., 2019*; *Homan et al., 2014*) to assess *MAPT* pre-mRNA and mature mRNA structures in T47D cells, a breast cancer line, and in neuronal SH-SY5Y cells. We used region-specific primers (*Smola et al., 2015*) to selectively amplify mature 3R and 4R transcripts during library preparation (*Supplementary file 4*; Materials and methods). This approach leverages the read-through capability of MaP technology to probe the structure of distinct alternatively spliced isoforms in the same cells. High DMS reactivities correspond to less structured regions, whereas low DMS reactivities correspond to more structured regions. DMS reactivities for replicates and cell lines were highly correlated (*Figure 1—figure supplement 4A*; *Figure 1—figure supplement 4B*; *Figure 1—figure supplement 5A*; *Figure 1—figure supplement 5B*).

As an internal control for our probing experiments, we also collected DMS-MaP data for the small subunit ribosomal RNA (SSU) (*Figure 1—figure supplement 6*), which has a well-defined secondary structure (*Petrov et al., 2014*). As expected, the DMS reactivities of unpaired nucleotides were significantly higher than for paired nucleotides both for RNA probed in cells and for RNA isolated from cells prior to probing (*Figure 1—figure supplement 7A*). This experiment confirmed that our DMS probing recapitulates native RNA secondary structure regardless of the presence of proteins, consistent with previous studies (*Woods et al., 2017*; *Lackey et al., 2018*). We used the SSU in-cell reactivity data to calibrate the estimation of equilibrium ensembles (Materials and methods), and we confirmed that structure modeling guided by experimental DMS reactivities yielded a more accurate estimation of the SSU structure than the model not informed by chemical probing data (*Figure 1—figure supplement 7B*).

The median in-cell DMS reactivity of the mature *MAPT* isoforms was 0.22, significantly greater than the median in-cell DMS reactivity of the SSU, which was 0.008 (*Figure 1—figure supplement*

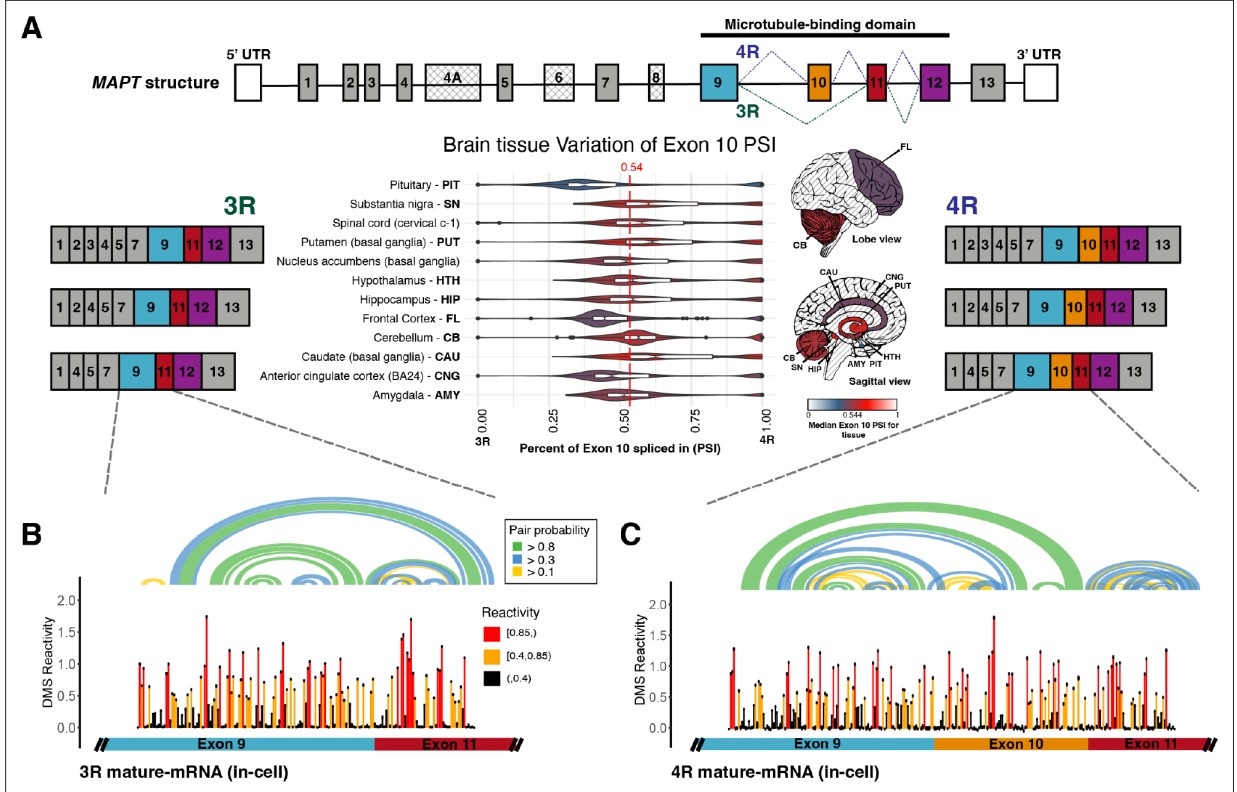

**Figure 1.** Three repeat (3R) and four repeat (4R) mature *MAPT* transcripts are expressed in a 1:1 ratio in samples from human subjects and mature exons appear to fold as independent structural units. (**A**) Ratio of 3R and 4R *MAPT* transcripts is approximately 1:1 among brain tissues. There are 14 exons alternatively spliced in *MAPT*. Exons 4 A, 6, and 8 are not found in brain mRNA. The four exons highlighted in color are repeat regions that form the microtubule binding domain in the Tau protein. Exon 10 is alternatively spliced to form the 3R or 4R isoform. This is highlighted by the alternate lines from the 5′ splice site of exon 9 to either the 3′ splice site of exon 10 (4R) or the 3′ splice site of exon 11 (3R). The six canonical transcripts found in the CNS can be separated into 3R and 4R transcripts. Percent spliced in (PSI) of exon 10 was calculated from RNA-seq data for 2315 tissue samples representing 12 CNS tissue types and collected from 375 individuals in genotype-tissue expression (GTEx) v8 database. The violin plot for each tissue type and the corresponding region on the brain diagram is colored by the median PSI for all samples of a given type. The patterned regions on the brain diagram indicate tissue types with no data. Tissue types spinal cord and nucleus accumbens are not visualized on the brain diagram. The median PSI of 0.54 among all tissue samples is indicated by the red dotted line. (**B**) In-cell dimethyl sulfate probing read out by mutational profiling (DMS-MaP) structure probing data across exon 9 – exon 11 junction of 3R mature *MAPT* transcript. T47D cells were treated with dimethyl sulfate (DMS). Structure probing data for junctions of interest were obtained using amplicon sequencing with region-specific primers (*Supplementary file 4*) following reverse transcription (RT) of extracted RNA. DMS reactivity is plotted for each nucleotide across spliced junctions. Each value is shown with its SE and colored by reactivity based on color scale. High DMS reactivities correspond to less structured regions, whereas low DMS reactivities correspond to more structured regions. The base pairs of the predicted secondary structure guided by DMS reactivities (using A/C nucleotides only) are shown in the arcs colored by pairing probabilities. (**C**) In-cell DMS-MaP structure probing data across exon 9 – exon 10 – exon 11 junction of 4R mature *MAPT* transcript.

The online version of this article includes the following source data and figure supplement(s) for figure 1:

**Source data 1.** Percent spliced in (PSI) of exon 10 plotted in *Figure 1A* and *Figure 1—figure supplement 1B*.

**Source data 2.** Dimethyl sulfate (DMS) reactivities plotted in *Figure 1B*.

**Source data 3.** Dimethyl sulfate (DMS) reactivities plotted in *Figure 1C*.

**Figure supplement 1.** Distribution of exon 10 percent spliced ins (PSIs) calculated for RNA-seq data from genotype-tissue expression (GTEx) database.

**Figure supplement 2.** Distribution of constitutively spliced exon 4 and alternatively spliced exon 2 percent spliced ins (PSIs) for 12 CNS, muscle-skeletal, colon-sigmoid, and breast-mammary tissue types.

**Figure supplement 3.** Distribution of gene expression levels for RNA-binding proteins (RBPs) implicated in the splicing regulation of *MAPT* exon 10.

**Figure supplement 4.** Dimethyl sulfate (DMS) structure probing data for mature *MAPT* three repeat (3R) and four repeat (4R) isoforms.

**Figure supplement 5.** In-cell dimethyl sulfate (DMS) reactivity data from T47D and SH-SY5Y.

**Figure supplement 6.** Dimethyl sulfate (DMS) structure probing data for small subunit (SSU) collected from T47D cells.

**Figure supplement 7.** Comparing dimethyl sulfate (DMS) structure probing data for small subunit (SSU) vs. *MAPT*.

**Figure supplement 8.** Scatter plot showing the percentage of intra-exon base pairs out of total number of base pairs.

*7C*). This difference was recapitulated in cell-free samples (*Figure 1—figure supplement 7C*). These results suggested that the nucleotides of the mature *MAPT* isoforms were more accessible and less paired overall as compared with the highly structured SSU. Reactivities of exon 9 and exon 11 were highly correlated between the 3R and 4R isoforms (*Figure 1B and C*; *Figure 1—figure supplement 4C*). In the 4R isoform, approximately 89% of base pairs were contained within the exon units; only 11% of base pairs were between residues from exon 10 with those of exon 11 (*Figure 1—figure supplement 8*). This result suggests that the mature exons fold as independent structural units.

## *MAPT* pre-mRNA exon 10 – intron 10 junction is more structured than the mature isoforms in cells

RNA structure around exon-intron junctions has been shown to regulate alternative splicing (*Warf and Berglund, 2010*; *Buratti and Baralle, 2004*), and a hairpin structure at the exon 10 – intron 10 junction of *MAPT* pre-mRNA is implicated in establishing the 3R to 4R 1:1 isoform ratio (*Hutton et al., 1998*; *Varani et al., 1999*; *Grover et al., 1999*; *Donahue et al., 2006*). The structure of the *MAPT* pre-mRNA in the exon 10 – intron 10 junction region has been studied using biophysical techniques and chemical probing of in vitro-transcribed fragments and using computational methods (*Varani et al., 1999*; *Lisowiec et al., 2015*; *Tan et al., 2019*; *Chen et al., 2019*), but the pre-mRNA structure had not previously been analyzed in cells. We obtained DMS-MaP data for this junction from endogenous pre-mRNA in T47D cells (*Figure 2A*). Replicates were highly correlated (*Figure 2—figure supplement 1A*). Similar reactivity data were also observed in SH-SY5Y cells (*Figure 2—figure supplement 1C*), despite the likely differences in RBP populations compared to T47D cells (*Figure 1—figure supplement 3*).

The reactivities for exon 10 in the pre-mRNA and mature 4R isoform were highly correlated (*Figure 2—figure supplement 1B*). This high correlation was unexpected given that the pre-mRNA undergoes splicing during the 5-min treatment of the cells with DMS. As we observed for the mature 4R isoform, exon 10 in the pre-mRNA mostly formed base pairs with other exon 10 nucleotides (*Figure 2—figure supplement 2*). However, when we compared DMS reactivities for pre-mRNA and the mature 4R isoform, we found that DMS reactivity in exon 10 was significantly lower for the pre-mRNA (median in-cell DMS reactivity: 0.08) than for the 4R isoform (median in-cell DMS reactivity: 0.22) (*Figure 2—figure supplement 3*). This was also the case for RNA probed under cell-free conditions (*Figure 2—figure supplement 1D*). The pre-mRNA is thus apparently more structured than mature mRNA independent of protein protection.

## Disease mutations change the *MAPT* pre-mRNA structural ensemble and splicing of exon 10

Many RNAs adopt an ensemble of structures instead of a single structure (*Halvorsen et al., 2010*; *Adivarahan et al., 2018*). We posited that a structural ensemble near the *MAPT* exon 10 – intron 10 junction regulates exon 10 splicing and that disease-associated mutations alter the composition of the structural ensemble to disrupt splicing regulation. We used Boltzmann sampling of RNA structures supported by DMS reactivity data (*Spasic et al., 2018*) (Materials and methods) to sample 1000 structures for the wildtype (WT). We also generated ensembles for two RNAs that bear mutations in intron 10 that are known to alter *MAPT* splicing: (i) an A to C mutation at position +15 (+15A>C) that favors 3R isoform, and (ii) a C to G mutation at position +19 (+19C>G) that favors the 4R isoform (*Tan et al., 2019*). These mutant ensembles were generated using the same DMS reactivities as the WT RNA, with the exception of the mutation site (see Materials and methods), and thus provide a well-controlled prediction of the impact that each mutation will have on the ensemble.

We visualized the structural ensemble for the 3000 structures using t-distributed stochastic neighbor embedding (t-SNE) (*Maaten and Hinton, 2008*) and identified five clusters (*Figure 2B*; Materials and methods). Each dot in the t-SNE plot (*Figure 2B*) corresponds to a single structure and is colored by the $\Delta G^{\ddagger}$ of unfolding (*Mustoe et al., 2018a*) of the 5' splice site (*Figure 2—figure supplement 4A*), defined as the last three nucleotides of exon 10 and the first six nucleotides of intron 10 (*Yeo and Burge, 2004*). The $\Delta G^{\ddagger}$ is the cost of disrupting a given structure without allowing the RNA to refold (*Mustoe et al., 2018a*; *Mustoe et al., 2018b*). We quantified and visualized the density of structures from the t-SNE plot (*Figure 2C*) and calculated representative structures for each cluster (*Figure 2D* and *Figure 2—figure supplement 4B*; Materials and methods). The WT sequence forms

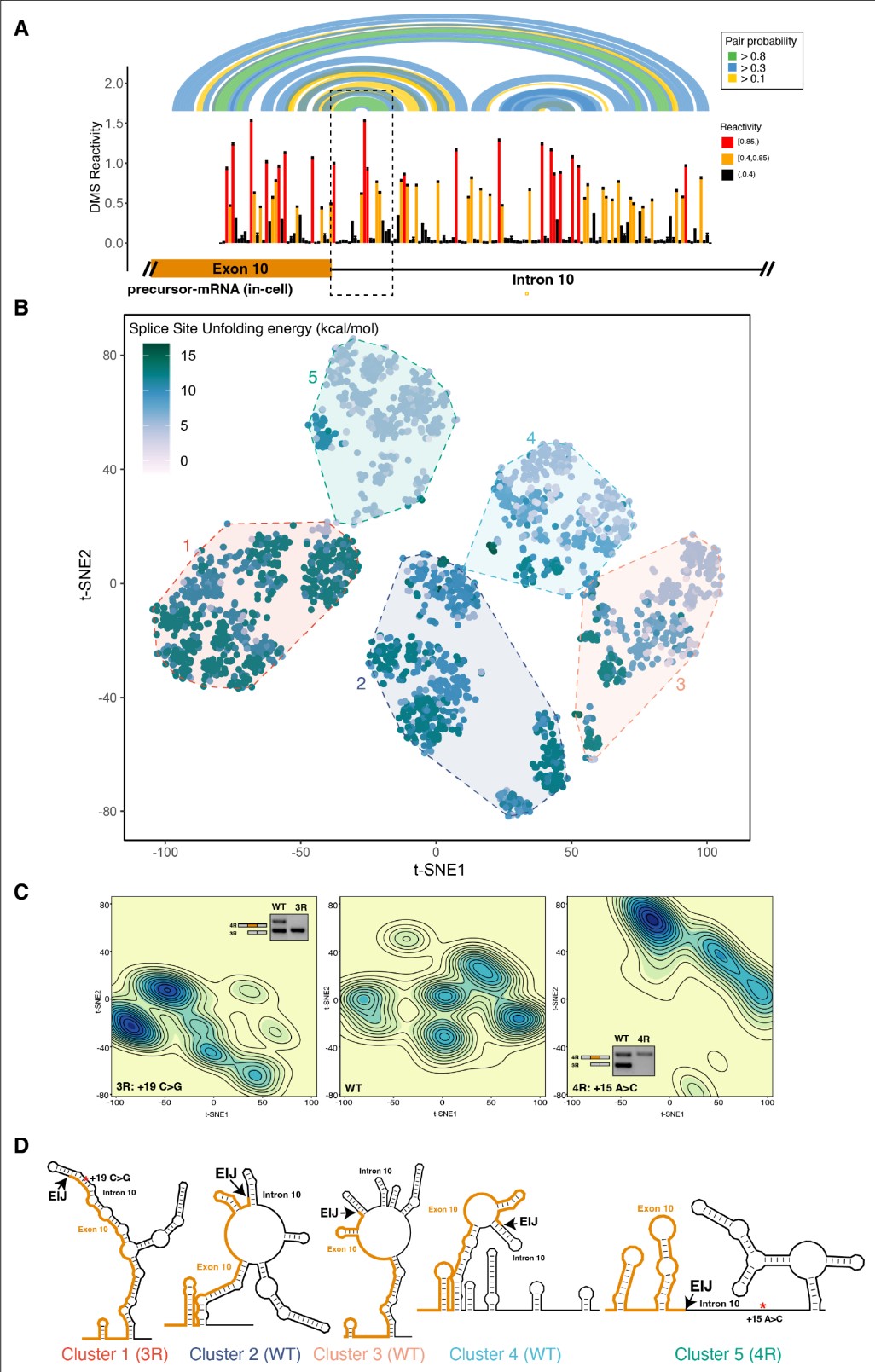

**Figure 2.** The four repeat (4R) and three repeat (3R) mutations shift dimethyl sulfate (DMS) reactivity-guided structural ensemble of exon 10 – intron 10 junction to be less structured and more structured, respectively. (**A**) In-cell dimethyl sulfate probing read out by mutational profiling (DMS-MaP) structure probing data across exon 10 – intron 10 junction of precursor *MAPT* transcript in T47D cells. Structure probing data for junctions of interest

*Figure 2 continued on next page*

*Figure 2 continued*

were obtained using amplicon sequencing with primers (*Supplementary file 4*) following RT of extracted RNA. DMS reactivity is plotted for each nucleotide. Each value is shown with its SE and colored by reactivity based on the color scale. High median DMS reactivities correspond to less structured regions, whereas low median DMS reactivities correspond to more structured regions. Base pairs of predicted secondary structure guided by A/C DMS reactivities are shown by arcs colored by pairing probabilities. Strongly predicted hairpin structure near exon-intron junction is highlighted by dotted box. (**B**) t-Distributed stochastic neighbor embedding (t-SNE) visualization of structural ensemble of wildtype (WT) and, +19C>G (3R) and +15A>C (4R) mutations. Structures were predicted using Boltzmann suboptimal sampling and guided by DMS reactivity data for A/C nucleotides generated in A. Data were visualized using t-SNE. Shown are 3000 structures corresponding to 1000 structures per sequence. Each dot represents a single structure and is colored by the calculated unfolding free energy of the 5' splice site at exon-intron junction (three exonic and six intronic bases). Clusters have been circled and enumerated using k-means clustering with k=5. (**C**) Density contour plots of structural ensemble of WT and, 3R, and 4R mutations. Contour plots were derived from the distribution of points on the t-SNE plot in B. The darker the blue, the higher the density of structures. Contour lines additionally represent density of points. Color scales for the three plots are identical. Inserts are gel images representative of splicing assays using a reporter plasmid expressing either the WT sequence, the +19C>G (3R) mutation or +15A>C (4R) mutation in HEK293 cells, where the RNA was extracted and reverse transcribed to measure the isoform ratio using specific PCR amplification (Materials and methods). In WT, both 3R (exon 9 – exon 11) and 4R (exon 9 – exon 10 – exon 11) isoforms are expressed (two bands). In the presence of the 3R mutation, only the 3R isoform is expressed (one band), whereas for the 4R mutation, only the 4R isoform is expressed (one band). Gel insets for the 3R and 4R mutation are in their respective density plots. (**D**) Representative structures for the five clusters are shown. The cluster number is indicated below each structure. The exon-intron junction is marked by EIJ on each structure. Positions of 3R and 4R mutations are marked by a red asterisk on their respective representative structures.

The online version of this article includes the following source data and figure supplement(s) for figure 2:

**Source data 1.** Dimethyl sulfate (DMS) reactivities plotted in *Figure 2A*.

**Source data 2.** t-Distributed stochastic neighbor embedding (t-SNE) coordinates plotted in *Figure 2B*.

**Figure supplement 1.** Dimethyl sulfate (DMS) structure probing data for precursor *MAPT* exon 10 – intron 10 region.

**Figure supplement 2.** Scatter plot showing the percentage of intra-exon/intron base pairs out of total number of base pairs.

**Figure supplement 3.** Box plots of distribution of dimethyl sulfate (DMS) reactivities for small subunit (SSU), three repeat (3R) isoform, four repeat (4R) isoform, and precursor mRNA (pre-mRNA) for in-cell and cell-free conditions.

**Figure supplement 4.** Mutations shift the structural ensemble.

---

structures distributed across the entire space with about 70% of structures found in clusters 2, 3, and 4 (*Figure 2—figure supplement 4B*). By contrast, in the +19C>G mutant that strongly favors the 3R isoform (*Tan et al., 2019*), more than 55% of structures belong to cluster 1, which is defined by a fully base-paired 5' splice site (*Figure 2D*). Conversely, greater than 50% of structures in the ensemble of the +15A>C mutant (cluster 5), which shifts the isoform balance entirely to 4R (*Tan et al., 2019*), were characterized by lower $\Delta G^{\ddagger}$ of unfolding for the splice site region (*Figure 2B and C*). Correspondingly, the 5' splice site for the cluster 5 representative structure was less structured than that of cluster 1 (*Figure 2D*). Based on these results, we concluded that mutations shift the structural ensemble of the *MAPT* exon 10 – intron 10 junction, and these structural shifts correspondingly change exon 10 splicing.

## Unfolding mRNA within the spliceosome B^act complex footprint yields the best prediction of exon 10 splicing level

RNA structure has been shown to control alternative splicing by regulating accessibility of key regions to spliceosome components (*McManus and Graveley, 2011*; *Warf and Berglund, 2010*). The 5' splice site is the minimum region of RNA that must be accessible for base pairing with the U1 snRNA (*Blanchette and Chabot, 1997*; *Singh et al., 2007*). In our structural ensemble analysis of the *MAPT* exon 10 – intron 10 junction (*Figure 2*), we found that shifts in the unfolding energy of the 5' splice site in WT and mutant pre-mRNAs corresponded to changes in exon 10 inclusion levels. However, the splicing cycle, orchestrated by the spliceosome, traverses multiple stages to prepare the pre-mRNA

and catalyze the two-step splicing reaction (*Matera and Wang, 2014*; *Figure 3A*). The RNA itself adopts many conformations as different components of the spliceosome bind to it (*Zhang et al., 2019*). Hence, we hypothesized that more than just the 5' splice site nucleotides might need to unpair to facilitate the splicing reaction. We analyzed high-resolution cryo-EM structures of the human spliceosome at Pre-B (PDB ID: 6Q × 9), B (PDB ID: 5O9Z), Pre-B$^{act}$ (PDB ID: 7ABF), and B$^{act}$ (PDB ID: 5Z56) stages (*Charenton et al., 2019*; *Bertram et al., 2017*; *Townsend et al., 2020*; *Zhang et al., 2018*) to quantify the number of nucleotides around the 5' splice site associated with the spliceosome (Materials and methods). The number of pre-mRNA nucleotides, as observed in each structure, increased through the splicing cycle (*Figure 3A*).

To identify the spliceosome complex footprint that best predicts splicing outcome, we examined the relationship between unfolding energy and splicing outcome for 20 synonymous or intronic mutations in exon 10 and intron 10 (*Figure 3—figure supplement 1A*). These mutations are more likely to affect splicing (*Supek et al., 2014*; *Lin et al., 2019*) and structure (*Sharma et al., 2019*; *Lin et al., 2016*) than mutations that alter the protein sequence. The distribution of $\Delta G^{\ddagger}$ of unfolding of the 5' splice site in the presence of these mutations was correlated with exon 10 PSI (*Figure 3—figure supplement 1B*). We then calculated the $\Delta G^{\ddagger}$ of unfolding of the RNA for regions overlapping the 5' splice site that correspond to the footprints of each of the four spliceosome intermediates. Features of the unfolding $\Delta G^{\ddagger}$ distribution, including mean and SD, were then used in a β-regression to predict exon 10 PSI (Materials and methods; *Equation 2*). Unfolding larger regions around the 5' splice site improved the predictive power of the model, and the B$^{act}$ complex footprint yielded the best prediction accuracy (R$^2$=0.89; *Figure 3B*). Crucially, we found that using features of the distribution of unfolding $\Delta G^{\ddagger}$ in the structural ensemble produced greater predictive power than simply using the unfolding $\Delta G^{\ddagger}$ of a single minimum free energy structure, supporting the importance of RNA ensemble behavior to splicing outcome (*Figure 3—figure supplement 1C*). We performed bootstrapping cross-validation and confirmed that unfolding the RNA within the B$^{act}$ spliceosome complex yielded the best prediction (*Figure 3C*). Synonymous mutations that alter exon 10 inclusion lie a mean distance of 54 nucleotides from the exon-intron junction, whereas those in the intron are a mean of 14 nucleotides from the junction. The variation in bootstrapped correlation coefficients decreased as a larger region around the exon-intron junction was unfolded, suggesting that the synonymous mutations affect distal structures.

We then tested the structural ensemble-based model on an additional 24 non-synonymous and compensatory mutations found in exon 10 and intron 10. Compensatory mutations are double mutations that were designed to rescue changes in exon 10 splicing caused by a single mutation (*Grover et al., 1999*). Although the model performed well for compensatory mutations (median bootstrapped R$^2$=0.76), it yielded significantly less accurate predictions for non-synonymous mutations (median bootstrapped R$^2$=−0.21) (*Figure 3—figure supplement 1D, E*). One clear limitation of this structure-only model is that it does not account for the effects of mutations on potential splicing regulatory elements (SREs) in the sequence, which are also known to control alternative splicing (*Wang and Burge, 2008*).

## Consideration of motif strengths of SREs improves prediction of exon 10 PSI for non-synonymous mutations

Exon 10 splicing is highly regulated by differential binding of RBPs to *cis*-SREs within exon 10 and intron 10 (*Qian and Liu, 2014*). The expression patterns of RBPs known to bind *MAPT* pre-mRNA vary across tissues and individuals (*Figure 1—figure supplement 3*) and are not predictive of exon 10 PSI. Additionally, while our structure-only model performs moderately well for 47 mutations (R$^2$=0.74) (*Figure 4—figure supplement 1A*, see *Supplementary file 2* for further details about mutations), the structure-only model performs particularly poorly for non-synonymous mutations (median bootstrapped R$^2$=−0.21, *Figure 4—figure supplement 1A*). Hence, we hypothesized that consideration of mutation-induced changes in binding of SREs might improve our model. We identified SREs by similarity to reported general enhancer and silencer hexamer motifs (*Fairbrother et al., 2002*; *Wang et al., 2004a*; *Wang et al., 2012*; *Wang et al., 2013*) (Materials and methods) and calculated changes to splice site, enhancer, and silencer motif strengths due to mutations (*Figure 4A*; Materials and methods). We found that using splice site strength as the sole predictor yielded poor prediction of exon 10 PSI for all mutation categories (*Figure 4B*; *Equation 4*). There was a weak positive correlation

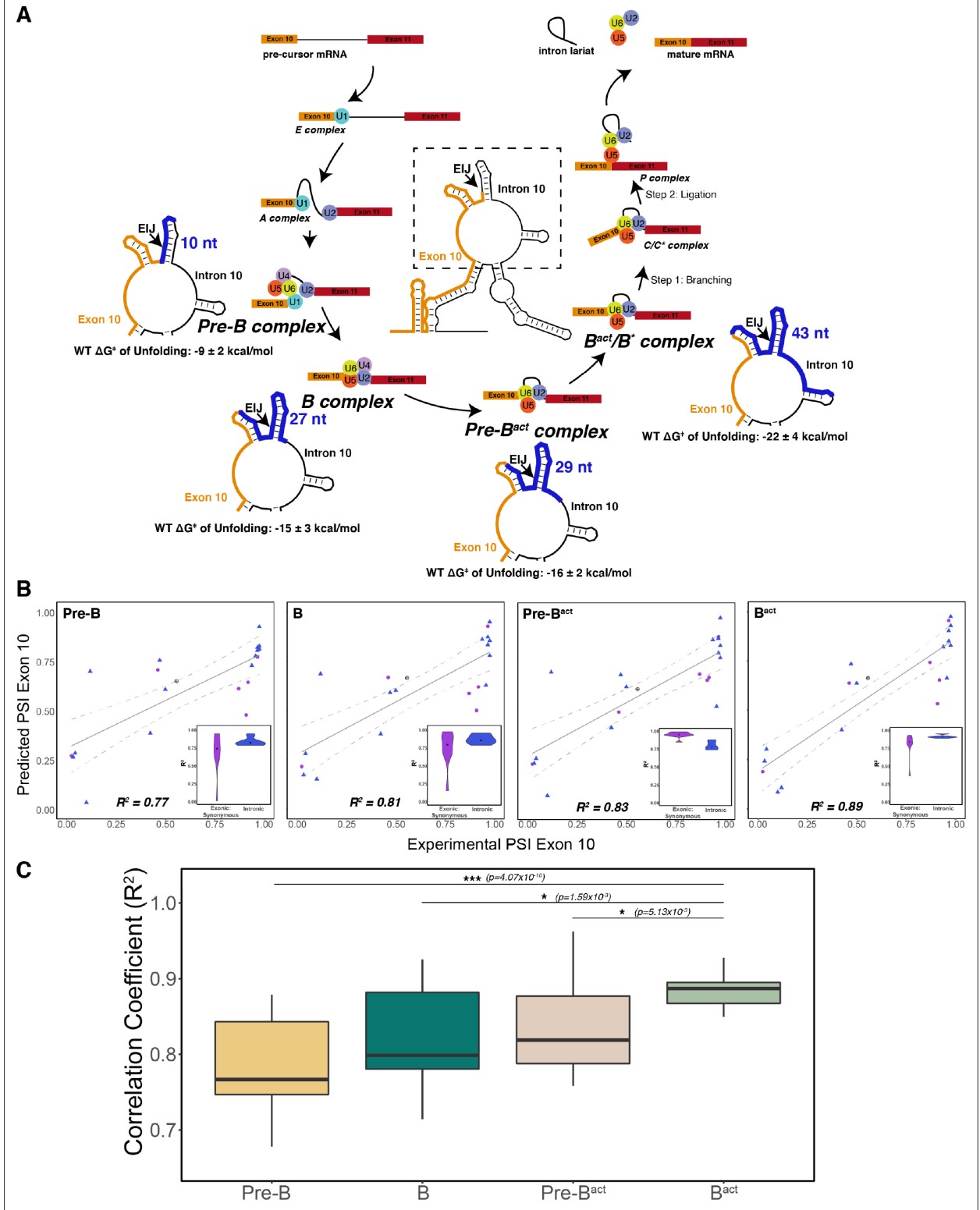

**Figure 3.** The best predictor of exon 10 percent spliced in (PSI) for intronic and synonymous mutations was the unfolding free energy of precursor mRNA (pre-mRNA) during the $B^{act}$ stage of splicing. (**A**) Spliceosome footprint on pre-mRNA during splicing cycle. Structure in the center of the cycle is the wildtype (WT) representative structure from ***Figure 2B***. The dotted box indicates the zoomed-in region at each stage of interest. Cryo-EM structures of the human spliceosome complex at stages Pre-B (PDB ID: 6Q × 9), B (PDB ID: 5O9Z), Pre-$B^{act}$ (PDB ID: 7ABF), and $B^{act}$ (PDB ID: 5Z56) are available in the Protein Data Bank. The region around the 5' splice site of pre-mRNA within the spliceosome at each stage is highlighted in blue on the zoomed-in

*Figure 3 continued on next page*

*Figure 3 continued*

representative structure. The number of nucleotides for each stage is as follows: Pre-B (2 exonic and 8 intronic); B (10 exonic and 17 intronic); Pre-B$^{act}$ (9 exonic and 20 intronic); B$^{act}$ (12 exonic and 31 intronic). These values represent the minimum number of nucleotides required to be unfolded to be accessible to the spliceosome. The mean free energy and SE to unfold RNA within the spliceosome at each stage are calculated for the WT structural ensemble and indicated under the zoomed-in structure. (**B**) Exon 10 PSIs of synonymous and intronic mutations predicted with the unfolding free energy of pre-mRNA within the spliceosome in B, Pre-B, Pre-B$^{act}$, and B$^{act}$ stages vs. corresponding experimental PSIs measured in splicing assays. Exon 10 PSIs were predicted using *Equation 2*. Gray line represents the best fit with dotted lines indicating the 95% CI. Pearson correlation coefficients (R$^2$) of experimental to predicted PSIs were calculated for each stage. Violin plots (inset) show R$^2$s calculated for each mutation category by training and testing on subsets of all mutations by non-parametric bootstrapping; synonymous (n=6), intronic (n=14), and WT (n=1). (**C**) Overall R$^2$ calculated for experimental vs. predicted exon 10 PSIs by non-parametric bootstrapping of mutations. Subsets of mutations were randomly sampled 10 times, trained and tested using unfolding free energy of the exon-intron junction region of pre-mRNA within the spliceosome for the respective splicing stage. Pearson's R$^2$ was calculated by comparing predicted PSIs to experimental PSIs. A two-tailed Wilcoxon rank sum test was used to determine significance between B$^{act}$ complex and the other three complexes. Level of significance: \*\*\*p-value<10$^{-6}$, \*\*p-value<0.001, and \* p-value<0.01.

The online version of this article includes the following figure supplement(s) for figure 3:

**Figure supplement 1.** Structure is a poor predictor of exon 10 percent spliced in (PSI) for exonic non-synonymous mutations.

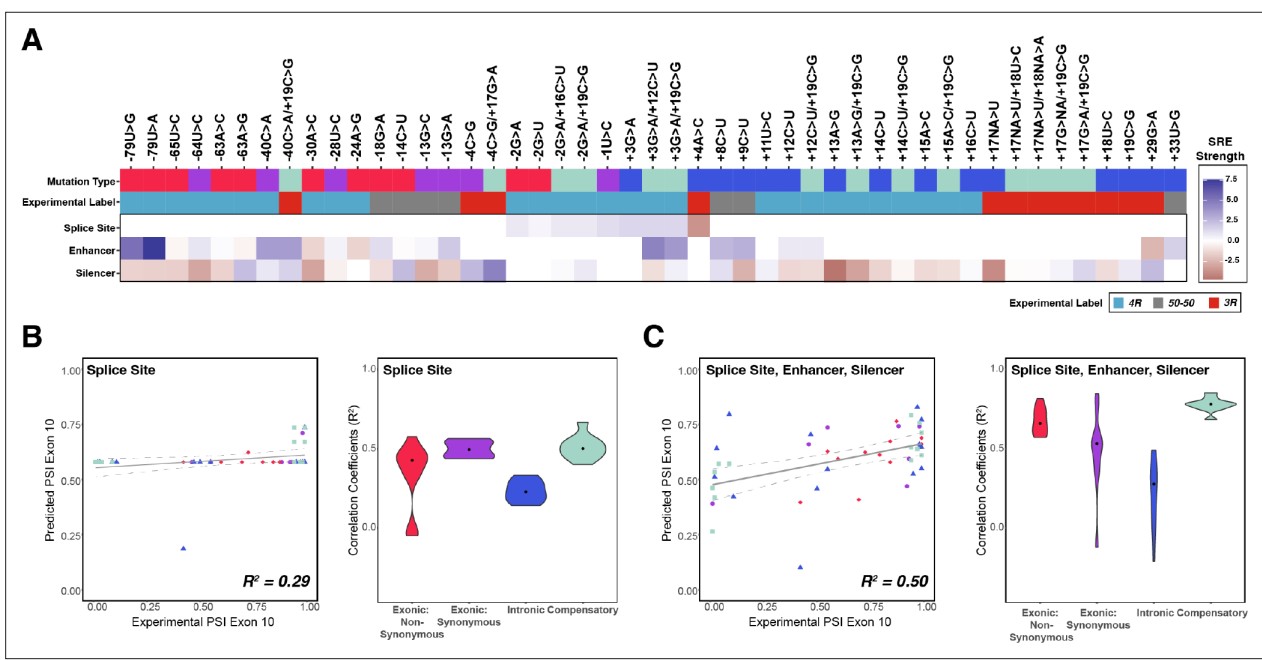

**Figure 4.** Combining the strength of all splicing regulatory elements (SREs) significantly improves prediction of exon 10 percent spliced in (PSI) compared to using only splice site strength. (**A**) Heatmap of SRE relative strength for 47 mutations compared with wildtype (WT). A value of 0 indicates mutation does not change WT SRE strength, positive values indicate SRE strength is greater than WT, and negative values indicate SRE strength is weaker than WT. Splice site strengths were calculated using MaxEntScan; a splice site was defined as the last three nucleotides of the exon and first six nucleotides of the intron. Enhancer and silencer strengths were calculated from position weight matrices of known motifs derived from cell-based screens (Materials and methods). Mutation type refers to whether the mutation is exonic non-synonymous, exonic synonymous, intronic, or compensatory. Experimental label is the label given by the original study that experimentally validated each mutation using a splicing assay. (**B**) Exon 10 PSIs of 47 mutations predicted from change in splice site strength and plotted against experimental PSIs measured in splicing assays. Exon 10 PSIs predicted using *Equation 4*. Each point on the scatterplot represents a mutation and is colored by mutation category. Gray line represents the best fit with dotted lines indicating the 95% CI. Pearson correlation coefficient (R$^2$) calculated of experimental to predicted PSIs. Violin plot shows R$^2$s calculated for each category by training and testing on subsets of all mutations by non-parametric bootstrapping; exonic non-synonymous (n=11), exonic synonymous (n=7), intronic (n=15), compensatory (n=14), and WT (n=1). (**C**) Exon 10 PSIs of 47 mutations predicted by combining change in splice site, enhancer, and silencer strength and plotted against experimental PSIs measured in splicing assays. Exon 10 PSIs predicted using *Equation 5*.

The online version of this article includes the following source data and figure supplement(s) for figure 4:

**Source data 1.** Values of heatmap plotted in *Figure 4A* which show the relative splicing regulatory strength for 47 mutations compared with wildtype.

**Figure supplement 1.** RNA-binding protein (RBP) binding motif strength is a poor predictor of exon 10 percent spliced in (PSI) for all mutations.

**Figure supplement 2.** RNA-binding protein (RBP) binding motif strength is a poor predictor of exon 10 percent spliced in (PSI) for all mutations.

between PSI and enhancer strength and a significant negative correlation between PSI and silencer strength (*Figure 4* and *Figure 4—figure supplement 1B*). When exon 10 PSI was modeled with the changes to the motif strength of all SREs, prediction accuracy increased ($R^2$=0.51; *Figure 4C*) compared with that obtained when only splice site strength was considered ($R^2$=0.29); for non-synonymous mutations accuracy was even higher ($R^2$=0.75).

Many RBPs have been identified that regulate *MAPT* exon 10 splicing (*Qian et al., 2011*; *D'Souza and Schellenberg, 2006*; *Kondo et al., 2004*; *Wang et al., 2004a*; *Gao et al., 2007*; *Ding et al., 2012*; *Broderick et al., 2004*; *Wang et al., 2010*; *Kar et al., 2006*; *Kar et al., 2011*; *Ray et al., 2011*). To determine whether focusing on binding motifs for these proteins would improve model accuracy, we identified RBP sites based on previous data from high-throughput sequencing of bound RNAs (*Dominguez et al., 2018*; *Ray et al., 2013*) (Materials and methods). Unlike SRE motifs, there was no clear pattern or correlation between motif strength changes due to *MAPT* mutations and exon 10 PSI (*Figure 4—figure supplement 2A, B*). Model prediction accuracy was lower ($R^2$=0.08, *Figure 4—figure supplement 2C*) than when predictions considered general SRE motifs. Thus, going forward, we chose to use SRE motifs for our combined models.

## Model with both RNA structure and SRE motif changes yields best prediction of exon 10 PSI

We next set out to determine if combining both structural and SRE features further improved prediction. Indeed, a combined interactive model consistently produced more accurate predictions of exon 10 PSI compared with a structure-only model and an SRE-only model for all mutation categories ($R^2$=0.89; *Figure 5A and B*). An alternative additive model had lower prediction accuracy ($R^2$=0.80) (*Figure 5—figure supplement 1A*), particularly for non-synonymous mutations (*Figure 5—figure supplement 1B*). This suggests that considering the category of mutation is critical in accurately modeling the effects on splicing.

To determine whether structure or SRE changes were responsible for the splicing phenotype of each individual mutant, we hierarchically clustered the four primary features (structure around 5' splice site, 5' splice site strength, enhancer strength, and silencer strength) for the 47 mutants that have been experimentally characterized (Materials and methods). Six categories emerged from the clustering of features (*Figure 5C*, and *Figure 5—figure supplement 1C*). For about 51% of mutations, both structure and SRE motif strength were altered in the same direction to either promote or inhibit exon 10 inclusion (*Figure 5D*). For the remaining mutations, structure and SRE strength changed in opposite directions. For 17% of mutants, structure dominated the direction of splicing. For about 23%, SRE strength was dominant (*Figure 5D*). Overall, these results support the conclusion that structure and SREs have equally important effects on regulation of splicing at this exon-intron junction.

## Mutations around the *MAPT* exon 10 – intron 10 junction skew to exon 10 inclusion

We next interrogated the model by performing a systematic in silico-mutagenesis analysis of the 100 nucleotides spanning the exon 10 – intron 10 junction (*Figure 6A*). Our model predicted that most mutations should result in inclusion of exon 10. This bias is consistent with the observation that about 75% of known disease-associated mutations in this region induce exon 10 inclusion (*Figure 6B* and *Figure 6—figure supplement 1A*). We found that a significantly greater proportion of disease-inducing mutations (76.4%) results in changes to both structure and SRE compared with uncategorized mutations (36.0%) (*Figure 6C*). Thus, mutations that alter both structure and SREs have a greater likelihood of causing disease than mutations that alter only structure or only an SRE. Intriguingly, mutations overall caused a slight skew toward a more structured exon-intron junction that would be expected to decrease inclusion of exon 10 (*Figure 6A*, *Figure 6—figure supplement 1B*); however, these same mutations altered SRE strength in a manner that skewed toward increased inclusion of exon 10 (*Figure 6—figure supplement 1C*), indicating that SREs act to counter the effect of structural changes. Our modeling suggests that a complex balance of structure and RBP binding results in the observed 1:1 ratio of the 3R to 4R *MAPT* isoforms.

To assess the general applicability of our model beyond our mutation training set, we predicted exon 10 PSIs for 55 variants of unknown significance (VUSs) found in dot-bracket SNP (dbSNP) (see *Supplementary file 3* for further details of VUSs). VUSs are mutations observed in the human population but

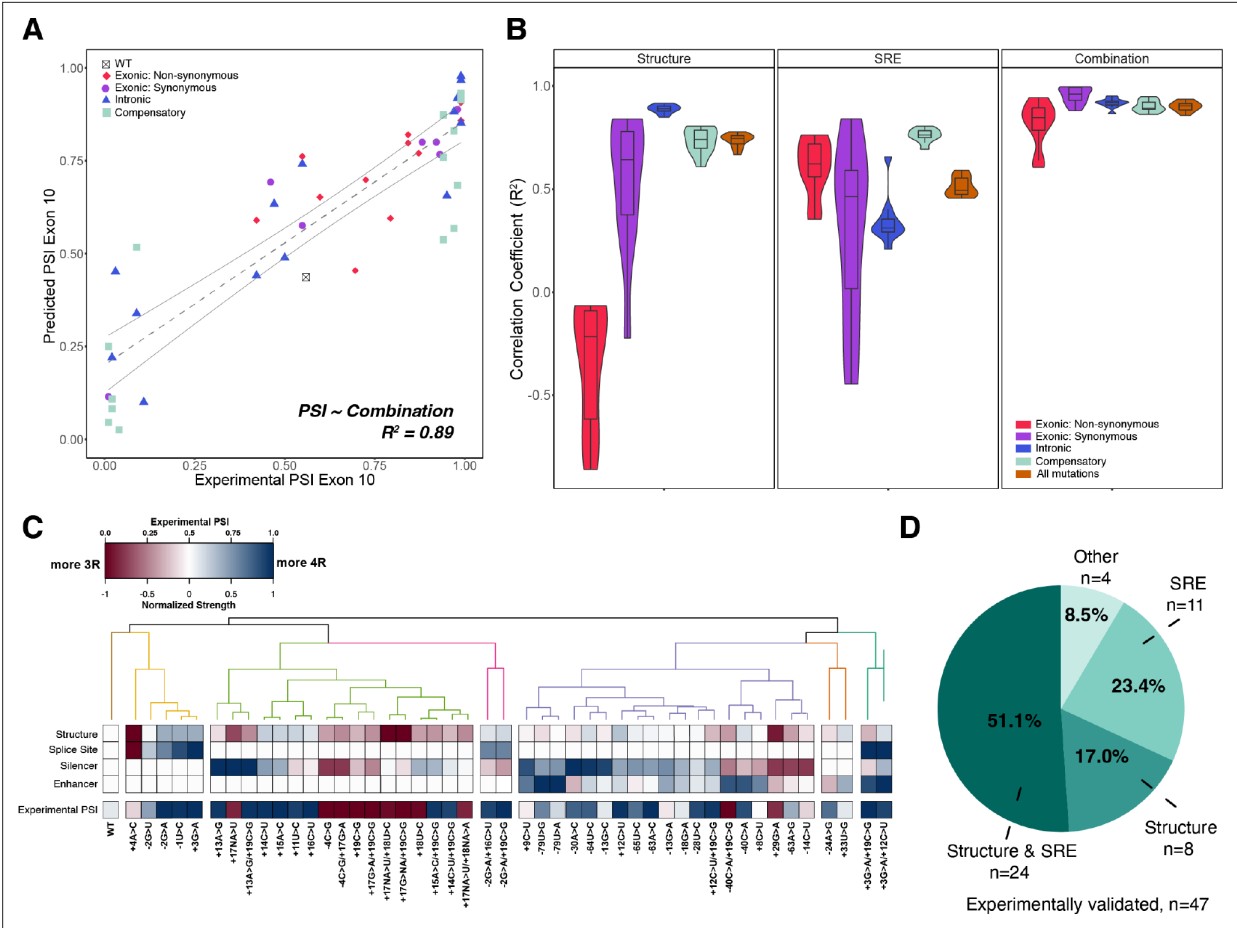

**Figure 5.** Combining structure and splicing regulatory element (SRE) strength into a unified model best predicts exon 10 percent spliced in (PSI). (**A**) Exon 10 PSIs of 47 mutations predicted from combined model using structure and SRE strength and fit to experimental PSIs measured in splicing assays. Exon 10 PSIs predicted using *Equation 7*. Each point on scatterplot represents a mutation and is colored by mutation category. Gray line represents the best fit with dotted lines indicating the 95% CI. Pearson correlation coefficient ($R^2$) calculated of experimental to predicted PSIs. (**B**) Violin plots of correlation coefficients for each mutation category for structure model, SRE model, and combined model. $R^2$s calculated for each mutation category by training and testing on subsets of all mutations by non-parametric bootstrapping 10 times. Structure model uses unfolding free energy of precursor mRNA (pre-mRNA) within spliceosome at B$^{act}$ stage as predictor. SRE strength model uses relative change in SRE strength as predictor. Combination model using both structure and SRE strength and weighs the features based on if mutation is intronic/synonymous or non-synonymous (Materials and methods). (**C**) Heatmap of the normalized changes in structure and SRE strength for each mutation clustered by affected features. Features were normalized such that a value of 1 predicts exon 10 spliced in (four repeat [4R] isoform, blue), whereas a value of 0 implies exon 10 spliced out (three repeat [3R] isoform, red). Mutations were clustered using hierarchical clustering on normalized features (Materials and methods). Experimental PSIs are plotted for each mutation with a PSI of 1 colored as blue, PSI of 0.5 colored as white, and PSI of 0 colored as red. (**D**) Pie chart showing the features that regulate exon 10 splicing for the 47 experimentally validated mutations. The pie chart was generated based on the heatmap in C. Exon 10 splicing for 51.1% of mutations is supported by changes in both structure and SRE. This implies that structure, at least one SRE motif, and PSI are all blue or all red in the heatmap in (C). Exon 10 splicing for 23.4% of mutations is supported by changes in SRE wherein one of the SREs is the same color as PSI. For 17.0% of mutations, structural changes support exon 10 splicing wherein structure and PSI are the same color. For 4 mutations (8.5%), the colors of none of the features match the color of PSI.

The online version of this article includes the following source data and figure supplement(s) for figure 5:

**Source data 1.** Values of heatmap plotted in *Figure 5C* which show the normalized changes in structure and splicing regulatory element (SRE) strength.

**Figure supplement 1.** Additive model of structure and splicing regulatory element (SRE) has poorer predictive performance than combined interactive model.

are not currently associated with disease. The mean exon 10 PSI for VUSs was 0.66, and 70% were within an SD of the mean (*Figure 6D*). We observed that only a few mutations were predicted to have a PSI of 0 (3R) (*Figure 6D* red bar). We therefore used splicing assays to experimentally determine the splicing preference of six instructive variants (Materials and methods): three VUSs predicted to be 3R,

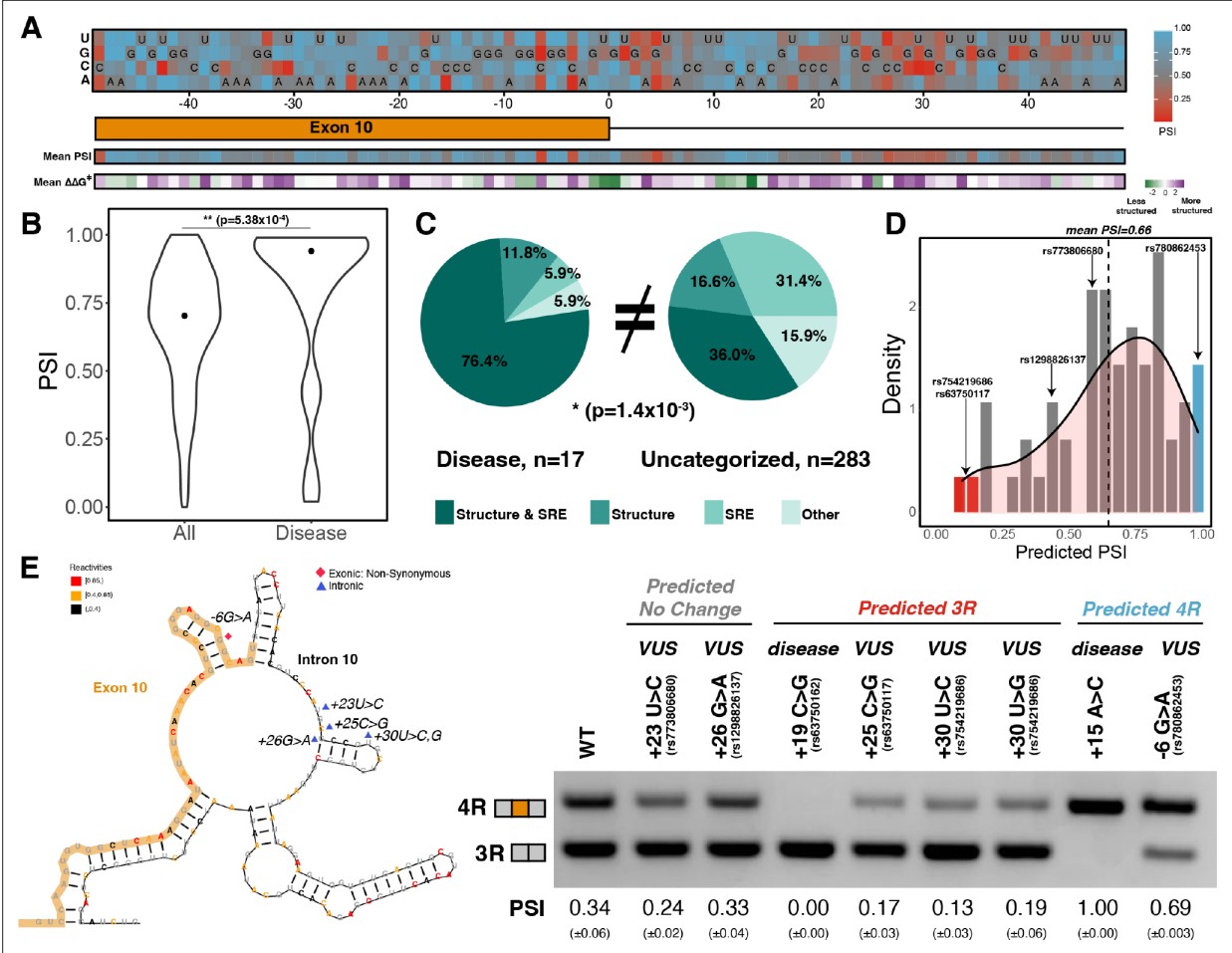

**Figure 6.** Combined model is predictive of exon 10 inclusion ratios for previously uncharacterized mutations. (**A**) Heatmap of predicted exon 10 percent spliced ins (PSIs) for every possible mutation around 100 nucleotide window of exon 10 – intron 10 junction. Combined model was trained using 47 mutations with experimental PSIs measured from splicing assays as shown in *Figure 5A* and then used to predict PSIs for all mutation combinations for 100 nucleotides around the junction. Tiles with sequence indicate the wildtype (WT) nucleotide at the position. Heatmap of mean PSI per position and mean relative change in unfolding free energy of pre-mRNA within spliceosome at $B^{act}$ stage compared with WT is shown below the gene diagram. (**B**) Violin plot of predicted PSIs for all possible mutations around exon 10 – intron 10 junction and only disease mutations. All possible mutations (n=300) and disease mutations (n=17). A two-tailed Wilcoxon rank sum test was used to determine significance between the two categories. Level of significance: \*\*\*p-value<$10^{-6}$, \*\*p-value<0.001, and \* p-value<0.01. (**C**) Pie chart showing features that drive exon 10 splicing for disease and presently uncategorized mutations. The pie chart was generated by quantifying the number of mutations for which the direction of predicted exon 10 PSI matched the direction of structure or splicing regulatory element (SRE) change. Exon 10 splicing for 76.4% of disease mutations is supported by changes to both structure and SRE compared with only 36.0% of uncategorized mutations. The difference in proportions was tested with a one-tailed Fisher's exact test. (**D**) Histogram displaying the distribution of predicted PSIs using the combined model for 55 variants of unknown significance (VUSs) found in dot-bracket SNP (dbSNP). Density curve was overlaid on top of histogram showing that predicted PSIs skew away from three repeat (3R). Dotted line shows mean predicted PSI of 0.66. VUSs tested in splicing assays are indicated by their dbSNP RS IDs. (**E**) Representative gel of RT-PCR data for splicing assay in the presence of VUSs. Splicing reporter was transfected into HEK293 cells. The mean exon 10 PSI displayed for each variant was calculated from three replicates, and SE is shown in brackets below. Structure diagram on left displays the location of the VUSs tested.

The online version of this article includes the following source data and figure supplement(s) for figure 6:

**Source data 1.** Values of heatmap of in-silico mutagenesis around exon 10 – intron 10 junction shown in upper panel of *Figure 6A*.

**Source data 2.** Values of heatmap of mean percent spliced in (PSI) predicted per position shown in lower panel of *Figure 6A*.

**Source data 3.** Values of heatmap of mean ΔΔG‡ per position shown in lower panel of *Figure 6A*.

**Source data 4.** Values of predicted percent spliced in (PSI) predicted for 55 variants of unknown significance (VUSs) shown in *Figure 6D*.

**Source data 5.** Band intensities of three replicates calculated for splicing reporter gels for six variants of unknown significance (VUSs) and wild type (WT) shown in *Figure 6E*.

**Figure supplement 1.** Effects of mutagenesis at each nucleotide in exon 10 – intron 10 region on percent spliced in (PSI).

one VUS predicted to be 4R, and two VUSs predicted to maintain the WT splicing ratio (*Figure 6D*). We found that all six predictions were correct (*Figure 6E*, *Figure 6—figure supplement 1D*). The three 3R VUSs caused the region around the exon-intron junction to become more structured, while the 4R VUS made this region less structured compared to WT (*Figure 6—figure supplement 1E*). SRE strength changes correctly predict exon 10 splicing direction for +30U>C and –6G>A (*Figure 6—figure supplement 1F*). For +23U>C and +26G>A, we observed changes in the degree of structure around the exon-intron junction and silencer strengths in diverging directions (*Figure 6—figure supplement 1E, F*) suggesting that these opposing changes preserve the WT 3R/4R ratio.

## Discussion

Splicing specificity is complex (*Baralle and Giudice, 2017*). The spliceosome does not rely on sequence alone to correctly identify 5' and 3' splice sites; other cues ensure correct binding to appropriate locations. The *MAPT* exon 10 – intron 10 junction is a well-studied example of the effect of 5' splice site secondary structure on splicing regulation. A hairpin was initially hypothesized to play a major role in splice site accessibility because disease mutations in this structure, close to the exon-intron junction, shifted the isoform balance to completely exclude or completely include exon 10 in the mature mRNA (*Hutton et al., 1998*; *Grover et al., 1999*). Nuclear magnetic resonance (NMR), cell-free chemical probing, and computation analyses confirmed the presence of the hairpin (*Varani et al., 1999*; *Chen et al., 2019*; *Lisowiec et al., 2015*). Recent studies have shown that structures determined in cell-free conditions can differ dramatically from those in cells (*Sun et al., 2019*; *Rouskin et al., 2014*). Our results suggest that this is not the case for the exon 10 – intron 10 junction region: in-cell chemical probing of the endogenous *MAPT* pre-mRNA provided strong evidence for formation of this hairpin in cells and for structural features not previously captured.

Our analysis also revealed that, in cells, exonic regions were less structured than introns, as also observed by *Sun et al., 2019*. Mature *MAPT* 3R and 4R are less structured in the region of exon 9 through exon 11 than is the pre-mRNA. The high correlations between structures of the exon in different *MAPT* isoforms and our finding that predicted exon 10 folding is only slightly impacted by the presence of intron 10 or exon 11 residues agree with previous observations of mRNAs. Specifically, mRNAs which encode yeast ribosomal proteins that indicate that RNA folding in both pre- and post-spliced exons is highly local and that most base pairs are intra-exon (*Zubradt et al., 2017*).

Unlike non-coding RNAs such as the ribosome and tRNA that rely on folding to a single, well-defined structure (*Petrov et al., 2014*), most RNAs are dynamic, unfolding, and refolding within a landscape (*Cruz and Westhof, 2009*; *Giegé et al., 2012*). We showed that structural ensembles have an important function at the exon 10 – intron 10 junction. If the 5' splice site was always paired, only the 3R isoform would be produced. However, the presence of 3R and 4R isoforms, usually in a 1:1 ratio, implies that the junction is accessible in a subset of the structures. We found disease-causing mutations produced distinct shifts in the ensemble of the *MAPT* exon 10 – intron 10 junction region; these shifts showed strong correlation with changes in the 3R to 4R isoform ratio and confirmed that ensembles are essential at this junction. Our ability to accurately predict the effects of mutations on ensembles significantly improved our quantitative model (*Figure 3—figure supplement 1C*).

The U1 snRNA base pairs with a nine nucleotide sequence around the exon-intron junction (*Roca et al., 2012*). However, our analysis of cryo-EM structures of the human spliceosomal assembly cycle revealed that a larger region of the pre-mRNA interacts with the spliceosome and must be unfolded during splicing. Our structural model performed most accurately when we required 43 nucleotides around the 5' exon-intron junction to be unfolded, corresponding to the region within the spliceosome B$^{act}$ complex. This observation suggests that a large region of the pre-mRNA is dynamically remodeled by the spliceosome and that structures distal to the exon-intron junction can regulate splicing. Our finding corroborates evidence that RNA structure near this exon-intron junction is extensive (*Tan et al., 2019*). Note that we do not claim that all 43 nucleotides need to remain fully unpaired during the splicing cycle, as the entire cycle is dynamic and likely involves other intermediate structures. Rather, our model argues that mRNA unfolding and accommodation into the B$^{act}$ complex are a key rate-limiting step in splicing, and considering this step is necessary to accurately model splicing outcome for a diverse set of mutations. Broadly, our definition of a functional footprint for splicing parallels a similar idea for translation initiation by the ribosome (*Corley et al., 2017*; *Mustoe et al., 2018a*; *Mustoe et al., 2018b*), for which the footprint is roughly 30 nucleotides. Thus, for both

translation initiation and splice site selection, there is a region in which RNA structure functions as a rheostat.

Considerable evidence supports a function for both SREs (and their corresponding RBPs) and RNA structure in controlling alternative splicing of exon 10 of *MAPT* (*Andreadis, 2012*). However, the relative importance of these two factors has been controversial. The regression model we developed clarifies that there is a cooperative relationship between RNA structure and SREs in driving splicing outcome. Exonic non-synonymous mutations promote splicing changes primarily by altering SRE motifs, whereas exonic synonymous and intronic mutations altered RNA structure. A combined model that accounted for both structure and SREs was the most accurate predictor of exon 10 PSI (*Figure 5D*). It was previously proposed that exon 10 is alternatively spliced due to a weak 5' splice site (*D'Souza and Schellenberg, 2005*), and indeed, we found that mutations that strengthened the splice site increase inclusion of exon 10 (*Figure 4A*). SRE strength alterations overall skewed more toward increased exon 10 inclusion, which suggest that SREs and the RBPs that bind them buffer the effects of RNA structure to maintain the 1:1 isoform ratio.

Although structure and SREs had opposing effects on splicing outcomes, disease variants frequently resulted in a synergistic effect on splicing outcome. The combined model was directly validated by accurate prediction of the effects of six previously untested VUSs on exon 10 splicing (*Figure 6E*). Few VUSs were predicted to completely exclude exon 10 from the mature mRNA. Only five VUSs had PSIs less than 0.25. Our model accurately predicted the effect of the three with the lowest predicted PSI. Our systematic computational mutagenesis revealed a hotspot for mutations around 25–30 nucleotides downstream of the exon-intron junction that were predicted to result in production of only the 3R isoform (*Figure 6A*). Indeed, the experimentally validated VUSs with PSIs less than 0.25 were in this region.

In principle, our splicing model can be extended to other exon-intron junctions, although RBPs that recognize SRE motifs have different binding contexts (*Dominguez et al., 2018*) and the exact binding preferences of the RBPs that regulate the junction of interest are currently unknown. Another limitation is that the current model does not consider structural and sequence features around the 3' splice site (in the case of *MAPT* exon 10, the intron 9 – exon 10 junction) that are expected to impact exon 10 splicing regulation. Although our model provides an exact PSI prediction for each mutation, we emphasize that its principal utility is in predicting the direction in which the 3R to 4R isoform ratio shifted from the WT ratio.

In brain tissue from healthy individuals, exon 10 PSIs varied between individuals and between tissues within an individual (*Figure 1A*). Even in individuals with progressive supranuclear palsy, a tauopathy in which *MAPT* variants are implicated, there was variability in exon 10 PSIs in different brain tissues (*Majounie et al., 2013*). Thus, although our model combines both structural and sequence features to achieve quantitative prediction accuracy of the 3R to 4R ratio for a wide range of disease mutations (synonymous, non-synonymous, intronic, and exonic), it is not clear that PSI alone is predictive of severity of disease for the broad class of tauopathies (*Majounie et al., 2013*). Disease severity is compounded by other factors including gene-gene interactions and environmental factors. As such, the value of our model stems more from how it incorporates RNA structure in predicting alternative splicing rather than as a direct predictor of disease severity. Many neurodegenerative diseases are caused by mutations around the *MAPT* exon 10 – intron 10 junction, and there are no approved therapeutics that target this junction. Our work suggests that it is crucial to consider the larger structural context of this region of the pre-mRNA and the interplay between structure and SREs when considering the consequences of mutations on splicing regulation and the design of potential therapeutics to alter this ratio.

# Materials and methods

**Key resources table**

| Reagent type (species) or resource | Designation | Source or reference | Identifiers | Additional information |
|---|---|---|---|---|
| Cell line (*Homo sapiens*) | T-47D | ATCC | HTB-133 | Epithelial cells from ductal carcinoma of breast |
| Cell line (*H. sapiens*) | SH-SY5Y | ATCC | CRL-2266 | Neuroblastoma cells from bone marrow |

*Continued on next page*

*Continued*

| Reagent type (species) or resource | Designation | Source or reference | Identifiers | Additional information |
| --- | --- | --- | --- | --- |
| Recombinant DNA reagent | Wildtype splicing reporter plasmid for *MAPT* exon 10 – intron 10 junction | *Tan et al., 2019* | | |
| Sequence-based reagent | Exon10Intron10-Fwd | This paper | PCR primers | AACGTCCAGTCCAAGTGTGG |
| Sequence-based reagent | Exon10Intron10-Rev | This paper | PCR primers | CATTCACCCAGAGGTCGCAG |
| Sequence-based reagent | 3 R/4R-Fwd | This paper | PCR primers | GCAGTGGTCCGTACTCCAC |
| Sequence-based reagent | 3 R/4R-Rev | This paper | PCR primers | TGATGGATGTTGCCTAATGAGC |

## Analyses of *MAPT* sequencing data for GTEx tissue types

Aligned BAM files of individual samples from the GTEx v8 project for tissue types with *MAPT* TPM greater than 10 were accessed in the AnVIL/Terra environment (*Kumar, 2020a*). Reads aligning to *MAPT* were extracted (*Kumar, 2020b*) and downloaded. Exons 2, 4, and 10 PSIs were quantified per BAM file with Outrigger (*Song et al., 2017*) using the *MAPT* transcriptome reference from Ensembl GRCh38. Only samples with at least 10 reads mapping across the exon-intron junction of interest were considered. For exon 10 PSI, median values for each tissue type were calculated and then visualized on the brain diagram with R package, CerebroViz (*Bahl et al., 2017*). Source file for *Figure 1* provides exon 10 PSI values for the 2962 samples. An ANOVA test was run in R to test significance in variation of exon 10 PSI between individuals vs. within an individual (for individuals with *MAPT* expression in more than seven tissues) (*Supplementary file 1*). TPMs for RBPs known to affect the splicing regulation of *MAPT* exon 10 were extracted, and their distributions in brain tissues were plotted using ggplot2.

## Culture of T47D and SH-SY5Y cells

Mammary gland carcinoma cells (T47D) were cultured in RPMI 1640 medium, supplemented with 10% fetal bovine serum (FBS) and 0.2 units/mL of human insulin at 37°C and 5% $CO_2$. Bone marrow neuroblastoma SH-SY5Y cells were cultured in 1:1 mixture of 1× minimum essential medium and 1× F12 medium, supplemented with 10% FBS at 37°C and 5% $CO_2$. Cell lines were obtained from the UNC Tissue Culture facility (TCF). UNC TCF together with Genetica Cell Line Services (a subsidiary of LabCorp) ensures stringent quantitative PCR-based mycoplasma contamination testing for cell or DNA samples. Genetica Cell Line Services also perform STR profiling of cell lines.

## In-cell DMS-MaP probing of *MAPT* RNA

Approximately 20 million T47D cells and 30 million SHSY-5Y cells were harvested by centrifugation and resuspended in 300 mM bicine, pH 8.3, 150 mM NaCl, and 5 mM $MgCl_2$ followed by treatment with DMS (1:10 ethanol diluted) for 5 min at 37°C as previously described (*Mustoe et al., 2019*). For the negative control (unmodified RNA) ethanol, instead of DMS, was added to cells. After incubation, the reactions were neutralized by addition of equal volume of ice cold 20% β-mercaptoethanol. Total RNA was extracted using Trizol (ThermoFisher Scientific), treated with TURBODNase (ThermoFisher Scientific), purified using Purelink RNA mini kit (ThermoFisher Scientific), and quantified based on absorbance determined with a NanoDrop spectrophotometer.

## Cell-free DMS-MaP probing for *MAPT* RNA

Approximately 10 million T47D cells in 10 cm plates were used. Growth media was removed, following which cells were trypsinzed (Tryple, ThermoFisher Scientific), and the pellet was washed with PBS. Total RNA was extracted by Trizol (ThermoFisher Scientific), chloroform, and isoamyl alcohol (24:1, Sigma-Aldrich) using phase lock heavy tubes (5PRIME Phase Lock Gel) followed by Purelink RNA mini kit purification (ThermoFisher Scientific) and on-column DNase digestion (PureLink DNase, ThermoFisher Scientific). RNA was quantified by NanoDrop spectrophotometer. 10 µg of RNA was resuspended in 90 µL of bicine buffer (200 mM Bicine pH 8.3, 100 mM NaCl, and 10 mM MgCl2) with 20 U of RNase inhibitor (NEB) and incubated at 37°C for 10 min. Samples were treated with 10 µL of DMS diluted in ethanol (1:10) for 5 min at 37°C. For the negative control (unmodified RNA), instead of DMS,

an equivalent amount of ethanol was added to the extracted RNA. After incubation, all reactions were neutralized by addition of 100 µL of ice cold 20% by volume β-mercaptoethanol and kept on ice for 5 min. Reaction by-products were removed by RNA purification with the Purelink RNA mini kit (ThermoFisher Scientific) before error-prone reverse transcription.

## DMS-MaP cDNA synthesis, library construction, and sequencing of *MAPT* RNA

Purified RNA (9 µg) was reverse transcribed using Random Primer 9 (NEB) and SuperScript II reverse transcriptase under MaP conditions as described previously (*Smola et al., 2015*). A no-reverse transcriptase control was also prepared. The resultant cDNA was purified over a G50 column (GE Healthcare) and subjected to second-strand synthesis (NEBNext Second Strand Synthesis Module). *Supplementary file 4* lists PCR primers used for library generation. The cDNA was amplified with the NEB Q5 HotStart polymerase. Secondary PCR was performed to introduce TrueSeq barcodes (*Smola et al., 2015*). All samples were purified using the Ampure XP beads (Beckman Coulter), and quantification of the libraries was performed with Qubit dsDNA HS Assay kit (ThermoFisher Scientific). Final libraries were run on Agilent Bioanalyzer for quality check. TrueSeq libraries were then sequenced as paired-end 2 × 151 and 2 × 301 read multiplex runs on MiSeq platform (Illumina) for pre-mRNA and mature mRNA, respectively. Sequenced reads have been uploaded to the NCBI SRA database under BioProject ID PRJNA762079 for in-cell data and PRJNA812003 for cell-free data.

## In-cell DMS-MaP probing of SSU

For in-cell rRNA structure data, approximately 10 million T47D cells were used for each condition. Growth media was removed, followed by addition of 1.8 mL of 200 mM bicine, pH 8.3, and treatment at 37°C with 200 µL of DMS diluted in ethanol (1.25% final DMS) for 5 min. For the negative control ethanol was added instead of DMS. After incubation, all reactions were neutralized by addition of equal volume ice cold 20% β-mercaptoethanol and kept on ice for 5 min. Total RNA was extracted using Trizol (ThermoFisher Scientific) and chloroform and isoamyl alcohol using phase lock heavy tubes (5PRIME Phase Lock Gel). RNA was purified using a Purelink RNA mini kit (ThermoFisher Scientific), treated with TURBODNase (ThermoFisher Scientific), and quantified.

## Cell-free DMS-MaP probing of SSU

Approximately 10 million T47D cells were trypsinzed (Tryple, ThermoFisher Scientific), and the pellet was washed with PBS. Total RNA was extracted using Trizol (ThermoFisher Scientific) and chloroform and isoamyl alcohol (24:1, Sigma-Aldrich) using phase lock heavy tubes (5PRIME Phase Lock Gel) followed by purification using a Purelink RNA mini kit purification (ThermoFisher Scientific) and on-column DNase digestion (PureLink DNase, ThermoFisher Scientific). RNA was quantified based on absorbance determined using NanoDrop spectrophotometer. For each sample, 10 µg of RNA was resuspended in 90 µL of 200 mM bicine, pH 8, 100 mM NaCl, and 10 mM $MgCl_2$ with 20 U of RNase inhibitor (NEB) and incubated at 37°C for 10 min. Samples were treated with 10 µL of DMS diluted in ethanol (1:10) for 5 min at 37°C. For the negative control, samples were treated with ethanol. After incubation, all reactions were neutralized by addition of 100 µL of ice cold 20% β-mercaptoethanol and kept on ice for 5 min. Reaction by-products were removed using a Purelink RNA mini kit (ThermoFisher Scientific) before error-prone reverse transcription.

## DMS-MaP cDNA synthesis, library construction, and sequencing of SSU

Purified RNA was reverse transcribed using Random Primer 9 (NEB) and SuperScript II reverse transcriptase under error prone conditions (*Smola et al., 2015*). The resultant cDNA was purified using G50 column (GE Healthcare) and subjected to second-strand synthesis (NEBNext Second Strand Synthesis Module). A standard Nextera DNA library protocol (Illumina) was used to fragment the cDNA and add sequencing barcodes. Samples were purified using Ampure XP beads (Beckman Coulter), and quantification of the libraries was performed with Qubit dsDNA HS Assay kit (ThermoFisher Scientific). Final libraries were run on Agilent Bioanalyzer for quality check. Gel purification (GeneJET, ThermoFisher Scientific) was performed as needed to remove primer dimer bands from libraries before sequencing. Libraries were sequenced as paired-end 2 × 151 read multiplex runs on MiSeq platform (Illumina).

Sequenced reads have been uploaded to the NCBI SRA database under BioProject ID PRJNA762079 for in-cell data and PRJNA812003 for cell-free data.

## DMS-MaP analysis

Sequenced reads were analyzed using the ShapeMapper pipeline (*Busan and Weeks, 2018*), version (v2.1.4). DMS probing data were collected for the exon 9 – exon 11 and exon 9 – exon 10 – exon 11 junctions using a single pair of primers listed in *Supplementary file 4*. The ShapeMapper pipeline ran for the two junctions in a single run with reference sequences for both junctions entered in one FASTA file. For the SSU, sequenced reads were first aligned to the SSU rRNA sequence using Bowtie2 parameters from *Busan and Weeks, 2018*. Using samtools, alignments with MAPQ score greater than 10 were kept, sorted, and converted back into FASTQ files after which the ShapeMapper pipeline was executed.

Per-nucleotide mutation rates were obtained from the profile file output by ShapeMapper. Raw DMS reactivities are computed as:

$$R_i = mutr_S - mutr_U$$

where $mutr_S$ is the mutation rate in the sample treated with DMS, and $mutr_U$ is the mutation rate in the untreated control. Raw reactivities were then normalized within a sample and per nucleotide type (A, C, U, and G). For each nucleotide type, reactivity rates were normalized by dividing the mean reactivity of the top 10% of reactivities after the most reactive 2% were removed (*Busan and Weeks, 2018*). Only As and Cs were used in structure modeling. For visualization of reactivity data, all four nucleotide types are shown in the reactivity figures and reported in the supplementary material, with G and U nucleotides normalized to a maximum value of 0.1.

## DMS reactivity-guided structure prediction

Previous versions of Rsample only utilized SHAPE parameters for calculation of the partition function. In order to use DMS data to guide secondary structure modeling by Rsample (*Spasic et al., 2018*; *Reuter and Mathews, 2010*), we used our SSU data to calibrate the expected relationship between DMS reactivity and base-pairing status. Since DMS primarily reacts with As and Cs, we only used reactivity data for these two nucleotides in all of our structure modeling. Using the SSU in-cell data and the known secondary structure (*Petrov et al., 2014*), we determined distributions for DMS reactivities for unpaired nucleotides, nucleotides paired at helix ends, and nucleotides paired in base pairs stacked between two other base pairs, which provide the sampling distributions needed for Rsample calculations (*Spasic et al., 2018*). These DMS data can be invoked by Rsample by the '--DMS' command line switch as part of RNAstructure 6.4 or later. The distributions had long tails to relatively high reactivities. We empirically found that limiting reactivities in the histograms and in the input data to a reactivity of 5 (where higher values are set to 5) gave the best performance at improving SSU secondary structure prediction. The '--max 5' parameter is used with Rsample to apply this limitation.

## Base-pairing probabilities for SSU

The partition function for the SSU was generated using Rsample, using either the sequence or using the sequence and the DMS reactivities. All possible *i-j* base-pairing probabilities were summed for each nucleotide *i* to generate a base-pairing probability per nucleotide *i*.

## ROC curves for predicting SSU base pairs

Using the known secondary structure of the SSU, we assigned a nucleotide as either 0 or 1 if it was paired or unpaired. Only As and Cs were considered. DMS reactivities were used to predict whether a nucleotide was paired; the higher the DMS reactivity, the more likely a nucleotide is unpaired. ROC curves and area under the curve values were generated using the plotROC (*Sachs, 2017*) R package.

## Minimum free energy and base-pairing probability modeling

Minimum free energy (MFE) and base-pairing probability 'arc' plots were generated using Superfold (*Siegfried et al., 2014*; *Smola et al., 2015*) modified to process DMS reactivity data. The original Superfold function used SHAPE parameters (*Deigan et al., 2009*) to fold an RNA sequence using the Fold and partition functions of the RNAstructure package. In our modified version of Superfold,

base pairing probabilities were computed using Rsample with DMS-specific parameters for A and C nucleotides. MFE structures were computed using Fold from RNAstructure with DMS reactivities input using the '--DMS' option. For structure modeling applications, G and U reactivities were set to –999 (no data).

### In silico co-transcriptional folding of exon 10 – exon 11 and exon 10 – intron 10 regions

We folded exon 10 using the modified Superfold function as described above, inputting a truncated DMS reactivity map file containing just reactivities of exon 10. We then added nucleotides from exon 11 to intron 10 one at a time and re-ran Superfold after each addition inputting the DMS reactivity map file modified to only the folded nucleotides. At every additional nucleotide, we calculated the number of base pairs within exon 10 and the total number of base pairs for the current sequence. We plotted the percentage of intra-exon/intron base pairs of total number of base pairs at every additional nucleotide.

### Generating a structural ensemble of the exon 10 – intron 10 region of *MAPT*

The partition function of the exon 10 – intron 10 region of *MAPT* for WT and mutants was calculated with DMS reactivities from the WT pre-mRNA as restraints using Rsample (*Spasic et al., 2018*). For modeling mutant sequences, DMS reactivities collected for the WT sequence were used to restrain the structural space with the reactivity at each mutation site set to –999. The program stochastic (*Reuter and Mathews, 2010*) was used to sample 1000 structures from the Boltzmann distribution wherein the likelihood structure is sampled was proportional to the probability that it occurred in the distribution (*Ding and Lawrence, 2003*).

### t-SNE visualization of structural ensembles

For each sequence, the 1000 structures in CT format for each ensemble were converted to dot-bracket (db) format with ct2dot (*Reuter and Mathews, 2010*), after which the db structure was transformed into the element format using rnaConvert in the Forgi package (*Kerpedjiev et al., 2015*). In the element format, every base is represented by the subtype of RNA structure in which it is found: stem (s), hairpin (h), loop (m), 5' end (f), and 3' end (t). Hence, each db structure is a string of characters. These characters were digitized (f, t:0, s:1, h:2, and m:3) to create a numerical matrix with 1000 rows and 234 columns, the length of the exon 10 – intron 11 region. Combining the matrices for the three sequences resulted in a 3000 × 234 matrix. This matrix was entered into the t-SNE function from the scikit-learn Python package (*Pedregosa et al., 2011*), and dimensionality was reduced to a 3000 × 2 matrix, which was then plotted with ggplot2 (*Wickham, 2016*) in R. The $\Delta G^{\ddagger}$ of unfolding of the splice site was calculated for each of the 3000 structures as described below. Source file for *Figure 3B* lists t-SNE reduced data with corresponding free energies.

### Identification of representative structures for clusters

The 3000 × 2 matrix obtained after t-SNE dimensionality reduction was clustered using k-means clustering with the k-means function from the scikit-learn Python package (*Pedregosa et al., 2011*). The value of k was set to 5 as determined visually. Boundaries for each cluster were marked and colored using the ggscatter function in the R ggpubr package. A custom Python script was used to deduce the representative structure for each cluster by first calculating the most common RNA structure subtype at each nucleotide. The structure in the ensemble that was most similar to the RNA structure with the most common subtypes at each position, was chosen as representative of that cluster.

### Visualizing density of structures in t-SNE plot

To evaluate density of structures in clusters, a meshgrid was created for the three matrices corresponding to WT, 3R, and 4R mutant structures using the meshgrid function of NumPy (*Harris et al., 2020*) with a 1000-point interpolation, which returns two-dimensional arrays that represent all the possible x-y coordinates for the three matrices. A Gaussian kernel was fit and evaluated for each 1000 × 2 matrix with SciPy gaussian_kde function (*Virtanen et al., 2020*) to smoothen over the meshgrid.

Contour lines were generated for the smoothed data with Matplotlib contour function (*Hunter, 2007*), and contourf was used to plot the data.

## Quantifying nucleotides around the 5' splice site in cryo-EM structure

The Protein Databank (PDB) files for Pre-B (PDB ID: 6Q × 9), B (PDB ID: 5O9Z), Pre-B$^{act}$ (PDB ID: 7ABF), and B$^{act}$ (PDB ID: 5Z56) complexes were downloaded from the PDB website. A custom Python script was used to extract pre-mRNA from each PDB file. The number of nucleotides were counted for mRNA found upstream and downstream of the 5' exon-intron junction. The result was visually confirmed by visualizing the PDB on PyMol.

## Calculating ΔG$^{‡}$ of unfolding of a region of interest

The ΔG$^{‡}$ of unfolding energies of regions of interest were calculated using a custom Python script. The non-equilibrium unfolding energy of the region, defined as the energy require to unfold a specific region without allowing refolding (*Mustoe et al., 2018a*) is defined as follows:

$$\Delta G^{‡} = \Delta G^{fold} - \Delta G^{unfold} \tag{1}$$

The ΔG of the original folded structure (ΔG$^{fold}$) was calculated with the efn2 program in RNAstructure (*Reuter and Mathews, 2010*). Next, the base pairs within a region of interest were made single stranded by setting the base pair column value to be 0 in the CT file. From this modified CT file, we evaluate the ΔG of the unfolded structure (ΔG$^{unfold}$) with efn2. This was done for every suboptimal structure in the Boltzmann ensemble. For example, to determine the ΔG$^{‡}$ of unfolding of the splice site, we made all nucleotides within the last three nucleotides of the exon and the first six nucleotides of the intron single stranded.

## Calculating changes in strength of splice site and SRE motifs

The strength of the WT splice site was calculated with MaxEntScan (*Yeo and Burge, 2004*). Strength was recalculated if mutations were located in the last three bases of exon 10 or first six bases of intron 10. WT strength was subtracted from the mutant strength. A value of 0 implied no change in splice site strength, positive values implied that a mutation made the splice site stronger, resulting in increased inclusion of exon 10, and negative values implied that a mutation made splice site weaker and decreased inclusion of exon 10.

Over-represented hexamers in cell-based screens of general ESEs and ISEs and ESSs and ISSs were obtained from previous reports (*Fairbrother et al., 2002*; *Wang et al., 2004b*, *Wang et al., 2012* and *Wang et al., 2012*). Position weight matrices (PWMs) of hexamers for each category, calculated as described (*Fairbrother et al., 2002*), are collated in *Supplementary file 5*. There were eight clusters of ESE motifs, seven of ESS motifs, seven of ISE motifs, and eight of ISS motifs; each cluster had an associated PWM. For each PWM, a threshold strength was found by taking the 95th percentile value of strength of all possible k-mers of PWM length. This threshold was used to determine whether there was a valid SRE motif at a particular position. The strength of the PWM motif was calculated across the exon-intron junction using a sliding window. The only windows that differed between the WT and mutants were around the location of the mutation, and only windows with strength above the threshold were considered. The WT strength was subtracted from the mutation strength for each window, and all windows were then summed to yield a Δstrength for every PWM per mutation. The average of the non-zero Δstrengths was calculated for ESE, ESS, ISE, and ISS categories. The ESE and ISE Δstrengths were summed to obtain an enhancer strength, and the ESS and ISS Δstrengths were summed to obtain a silencer strength. *Supplementary file 6* presents all SRE Δstrengths for the 47 mutations and 55 VUSs.

## Calculating the change in strength of RBP motifs

Previously determined position frequency matrices (PFMs) for SRSF1, SRSF2, SRSF7, SRSF9, SRSF10, PCBP2, RBM4, and SFPQ (*Ray et al., 2013*) were converted into PWMs by normalizing frequencies to 0.25 (the prior probability for nucleotide frequency) and calculating the log2 value. Position frequency matrices were calculated based on previously reported over-represented hexamers for SRSF11, SRSF4, SRSF5, and SRSF8 (*Dominguez et al., 2018*). PFMs for these RBPs were calculated as described previously (*Fairbrother et al., 2002*). Δstrength values for RBP motifs were calculated as

described for SRE motifs. The averages of non-zero values of RBPs implicated in either the inclusion or exclusion of exon 10 were computed separately. All RBP Δstrengths for the 47 mutations are listed in *Supplementary file 6*.

## Models and bootstrapping

Exon 10 PSI was limited to values between 0 and 1 with 0 signifying that no transcripts had exon 10 and 1 that all transcripts had exon 10. Hence, standard linear regression was not appropriate, and features were fit with a beta regression model to exon 10 PSI. Regression parameters were determined using the betareg package (*Cribari-Neto and Zeileis, 2010*) in R. Bootstrapping was performed by sampling without replacement 70% of the mutants. Pearson $R^2$ values between true values and predictions of the sample were calculated for the training set. This bootstrapping was executed 10 times resulting in a range of $R^2$s, ensuring that no subset of mutations skewed model performance. Since there were only four mutants that maintained the WT 1:1 3R to 4R ratio in our training set, we added three VUSs from dbSNP, which we experimentally verified preserved the WT splicing pattern (*Supplementary file 7*). The VUSs tested and added to the training set were assigned a PSI of 0.5 to indicate equivalence to the WT sequence. *Equation 2*, the structure ensemble model, uses four characteristics describing **X**, the $\Delta G^{\ddagger}$ of unfolding of the region of interest around the exon-intron junction for 1000 structures in the ensemble. The mean, SD, skew, and kurtosis were calculated for the $\Delta G^{\ddagger}$ values of 1000 structures in each ensemble. *Equation 3*, the minimum free energy model, uses just **Y**, the $\Delta G^{\ddagger}$ of unfolding of the exon-intron junction found within the spliceosome at the $B^{act}$ stage for the single minimum free-energy structure. *Equation 4*, the splice site model, uses the difference in splice site strength between WT sequence and a mutation where *SS* represents splice site. *Equation 5*, the combined SRE model, uses the difference in SRE strength between WT sequence and a mutation where *SS* represents splice site, *E* represents enhancer, and *S* represents silencer. *Equation 6*, the RBP model, uses the difference in RBP motif strength between WT sequence and a mutation where *Ex* represents RBPs involved in the exclusion of exon 10 and *In* represents RBPs involved in the inclusion of exon 10. *Equation 7* is the interactive model between structure and SRE, and *Equation 8* is the additive model. *isNonSynonymous*, *isSynonymous,* and *isIntronic* represent the category of mutation and is either 0 or 1. *Supplementary file 6* summarizes the performance of the models and features utilized.

$$PSI \sim Mean(X) + SD(X) + Skew(X) + Kurtosis(X) \tag{2}$$

$$PSI \sim Y \tag{3}$$

$$PSI \sim SS \tag{4}$$

$$PSI \sim E + S + SS \tag{5}$$

$$PSI \sim Ex + In \tag{6}$$

$$PSI \sim [Mean(X) + SD(X) + Skew(X) + Kurtosis(X)] * [isSynonymous + isIntronic] + [E + S + SS] * [isNonSynonymous] \tag{7}$$

$$PSI \sim [Mean(X) + SD(X) + Skew(X) + Kurtosis(X)] + [E + S + SS] \tag{8}$$

## Clustering of changes in structural and SRE features

For each feature, non-zero values greater than the 95th percentile value were set to the 95th percentile or, if less than the 5th percentile value, were set to the 5th percentile for visualization, after which all values were normalized to the maximum absolute value. Silencer Δstrength and mean $\Delta G^{\ddagger}$ of unfolding of ensembles were inverted to follow the visualization such that values closer to 1 would result in greater exon 10 inclusion and values closer to 0 would result in lower exon 10 inclusion. Features were then assigned values 0 or 1 depending on whether the feature changed at all in the presence of the mutation. These digitized features were clustered by hierarchical clustering resulting in six clusters. Each individual cluster was then clustered again by hierarchical clustering using the normalized feature values instead of values of 0 and 1.

## Splicing assays

HEK-293 cells (ATCC CRL-1573) were grown at 37°C in 5% $CO_2$ in Dulbecco's Modified Eagle Medium (Gibco) supplemented with 10% FBS (Omega Scientific) and 0.5% penicillin/streptomycin (Gibco). The WT splicing reporter plasmid was generously provided by the Roca lab (*Tan et al., 2019*).

Single-nucleotide point mutations were generated using a Q5 site-directed mutagenesis kit (NEB) and confirmed by Sanger sequencing or were custom ordered directly from GenScript. Reporter plasmids (2 µg) were transfected into HEK-293 cells in six-well plates when cells were 60–90% confluent using Lipofectamine 3000 (ThermoFisher Scientific). Cells were harvested after 1 day by aspirating the media. Cells were resuspended in 1 mL Trizol reagent (ThermoFisher Scientific). RNA was isolated using the PureLink RNA Isolation Kit (ThermoFisher Scientific) with on-column DNase treatment, following manufacturer's instructions. RNA (1 µg) was reverse transcribed to cDNA using Superscript VILO reverse transcriptase (ThermoFisher Scientific). Reverse transcriptions were performed by annealing (25°C, 10 min), extension (50°C, 10 min), and inactivation (85°C, 10 min) steps. Heat-inactivated controls were prepared by heating the reaction without RNA at 85°C for 10 min prior to adding RNA, then following the described reaction conditions. The cDNA was PCR amplified with NEB Q5 HotStart polymerase (NEB) using splicing assay primers from IDT (AGACCCAAGCTGGCTA GCGTT forward, GAGGCTGATCAGCGGGTTTAAAC reverse) with 25 cycles. PCR product was purified and concentrated using the PureLink PCR micro clean up kit (ThermoFisher Scientific) following manufacturer's instructions. Splicing products were visualized by loading ~200 ng of DNA on a 2% agarose gel in 1× tris-acetate EDTA buffer and staining with ethidium bromide. Gel images were quantified with ImageJ.

Supplementary files, figure source files, SNRNASMs, and code are available at GitHub repository: https://git.io/JuSW8, copy archived at swh:1:rev:4cb33b94bb8a864bc63fd5a3c96dae547914b20f; *Kumar, 2022*.

## Acknowledgements

This work was supported by the US National Institutes of Health R01 HL111527 and R35 GM 140844 to A.L., R01 GM076485 to D.M., and R35 GM122532 to K.M.W. A.M.M. is a CPRIT Scholar (RR190054). The authors wish to thank the Roca Lab for providing wild-type splicing reporter plasmids, Dr. Zefeng Wang for intronic splicing enhancer and silencer motifs, and Drs. Peter Castaldi and John Platig for insightful discussions.

## Additional information

### Competing interests

Kevin M Weeks: is an advisor to and holds equity in Ribometrix. The other authors declare that no competing interests exist.

### Funding

| Funder | Grant reference number | Author |
|---|---|---|
| National Institutes of Health | R01 HL111527 | Alain Laederach |
| National Institutes of Health | R35 GM 140844 | Alain Laederach |
| National Institutes of Health | R01 GM076485 | David H Mathews |
| National Institutes of Health | R35 GM122532 | Kevin M Weeks |
| Cancer Prevention and Research Institute of Texas | CPRIT Scholar (RR190054) | Anthony M Mustoe |
| National Institutes of Health | R35 GM142851 | Lela Lackey |

The funders had no role in study design, data collection and interpretation, or the decision to submit the work for publication.

## Author contributions
Jayashree Kumar, Conceptualization, Data curation, Formal analysis, Methodology, Software, Validation, Visualization, Writing – original draft, Writing – review and editing; Lela Lackey, Conceptualization, Funding acquisition, Methodology, Writing – original draft; Justin M Waldern, Data curation, Funding acquisition, Writing – original draft; Abhishek Dey, Funding acquisition, Writing – original draft; Anthony M Mustoe, Kevin M Weeks, Conceptualization, Writing – original draft; David H Mathews, Software, Writing – original draft; Alain Laederach, Conceptualization, Formal analysis, Methodology, Software, Visualization, Writing – original draft

## Author ORCIDs
Jayashree Kumar ⓘ http://orcid.org/0000-0001-6914-748X
Lela Lackey ⓘ http://orcid.org/0000-0003-2163-4005
Anthony M Mustoe ⓘ http://orcid.org/0000-0001-9346-1559
Alain Laederach ⓘ http://orcid.org/0000-0002-5088-9907

## Decision letter and Author response
Decision letter https://doi.org/10.7554/eLife.73888.sa1
Author response https://doi.org/10.7554/eLife.73888.sa2

## Additional files

### Supplementary files
• Supplementary file 1. ANOVA table for between individuals and within individuals Exon 10 PSI comparison.
• Supplementary file 2. Details on 47 experimentally tested *MAPT* mutations used in training model.
• Supplementary file 3. Details on 55 variants of unknown significance (VUSs) in *MAPT* from dbSNP.
• Supplementary file 4. Primers used for amplification of exon-exon or exon-intron junctions.
• Supplementary file 5. Re-calculated Position Weight Matrices for ESEs, ESSs, ISEs, ISSs.
• Supplementary file 6. Details on beta regression model results and features used for each training and test set.
• Supplementary file 7. Gel of RT-PCR data for splicing assay for new WT VUSs.
• Transparent reporting form
• Source data 1. Marked-up gel images for 3 replicates from MAPT splicing plasmid assay.
• Source data 2. Raw unmarked gel images for 3 replicates from MAPT splicing plasmid assay.

### Data availability
Sequencing data have been deposited in SRA under BioProject ID PRJNA762079 and PRJNA812003. DMS Reactivities are available as SNRNASMs at https://bit.ly/2WaDw6F. All data generated or analyzed during this study are included in the manuscript and supporting files; Source Data files have been provided for Figures 1,2,4,5 and 6. Modeling and feature generation code is uploaded at https://git.io/JuSW8.

The following datasets were generated:

| Author(s) | Year | Dataset title | Dataset URL | Database and Identifier |
|---|---|---|---|---|
| Kumar J, Lackey L, Waldern JM, Dey A, Mustoe AM, Weeks KM, Mathews DH, Laederach A | 2021 | MAPT mRNA in-cell chemical probing - sequenced reads | https://dataview.ncbi.nlm.nih.gov/object/PRJNA762079?reviewer=ccd0ed8huijlgqhd625453vr6f | NCBI BioProject, PRJNA762079 |
| Kumar J, Lackey L, Waldern JM, Dey A, Mustoe AM, Weeks KM, Mathews DH, Laederach A | 2021 | MAPT mRNA in-cell and cell-free chemical probing - DMS reactivities | https://bit.ly/2WaDw6F | SNRNASMs, 2WaDw6F |

*Continued on next page*

*Continued*

| Author(s) | Year | Dataset title | Dataset URL | Database and Identifier |
|---|---|---|---|---|
| Kumar J, Lackey L, Waldern JM, Dey A, Mustoe AM, Weeks KM, Mathews DH, Laederach A | 2021 | MAPT mRNA cell-free chemical probing - sequenced reads | https://dataview.ncbi.nlm.nih.gov/object/PRJNA812003?reviewer=bd798209t8o4gacqhs31rgctf6 | NCBI BioProject, PRJNA812003 |

The following previously published dataset was used:

| Author(s) | Year | Dataset title | Dataset URL | Database and Identifier |
|---|---|---|---|---|
| GTEx Consortium | 2017 | GTEx_Analysis_2017-06-05_v8_RNAseq_BAM_files | https://www.ncbi.nlm.nih.gov/projects/gap/cgi-bin/study.cgi?study_id=phs000424.v8.p2 | dbGAP, phs000424.v8.p2 |

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
