## [Editor Report]

This manuscript will be of interest to biologists who study RNA structure-function relationships in a broad range of systems, splicing researchers, and RNA structure bioinformaticians. An integrative analysis of RNA structure probing, model-based RNA folding energetics, cryo-EM data, and protein binding sequence motifs serves as the basis for a comprehensive, accurate, and robust framework for predictive models of splicing dynamics in a well-studied system. The modeling is leveraged by in silico mutagenesis that reveals novel insights into the mechanisms and tradeoffs that underlie the impact of disease-associated mutations on alternative splicing.

---

## [Decision Letter]

**Decision letter after peer review:**

Thank you for submitting your article "Quantitative prediction of variant effects on alternative splicing using endogenous pre-messenger RNA structure probing" for consideration by *eLife*. Your article has been reviewed by 3 peer reviewers, and the evaluation has been overseen by a Reviewing Editor and Naama Barkai as the Senior Editor. The reviewers have opted to remain anonymous.

Essential revisions:

1. The figure of SSU structure could be improved with clear nucleotide information along with reactivities. The ROC shows that the sequence seems more important than DMS reactivities. It might be better to quantitatively measure how many single-stranded nucleotides have high DMS reactivities. The DMS reactivities for unpaired As and Cs are much higher than those for paired As and Cs. Is it due to protein protection? It seems unreasonable to compare the reactivity profiles between SSU and MARPT isoforms since the profile difference might be only due the different interacted proteins.

2. It is interesting that 66% of base pairs were contained within exon units. Would the percentage remain similar when performing RNA folding with sliding window? If so, this observation could be explained as co-transcriptional folding.

3. The intron is more structured than the exon. Rather than RNA structure, is it possible that it is due to the protein protection? Would it remain a similar pattern under deproteinized conditions?

4. The unfolding free energy is based on the first step of 5'SS recognition (9 nts) for U1 snRNA. It is also possible that the RNA structure only affects U5 snRNA binding? The authors could test unfolding 3nt and see whether it will be similar or no effect. Again, will the clusters remain similar if the folding windows change?

5. It is not necessary to unfold all the nucleotides at one time. At the different steps during the splicing, the RNA structure might be very dynamic and remodelled in the process.

6. For the training set, is there any reason for using 20 synonymous and intronic mutations? A detailed clarification is necessary. Why not non-synonymous mutations?

7. The prediction could be improved by taking into account RBPs. The different DMS reactivity profiles are likely due to RBP binding. See the comment above.

8. Figure 6E. The first mutation the model predicted as "No change" seems to change; the splicing ratio does change at the leftmost column, although to a lesser extent than in the other columns called as "changed".

9. I would encourage the authors to come up with an explanation/hypothesis of why some mutations are more impactful, with respect to disease, than others.

10. Throughout the text, the authors discuss RNA conformations as being "more" or "less" structured and then "structured" or "unstructured." It would be beneficial to describe what it means to be "structured" versus "non-structured" and if they are truly the extremes or another way of saying more of less structured. It is difficult to follow as the text flips between these four ways to describe the structure, and then incorporates the direct measurement of the energy of splice site unfolding.

11. It would be good to include "MAPT" in the title considering all data was collected for this gene. If the author wanted to present this framework as a general model for splicing prediction, analysis of a second exon-intron junction would be required.

12. Results, first subsection: MAPT's PSI variability was analyzed in the context of different individuals and different cell types/tissues. This is followed by a statement that this isoform ratio is consistent. It would be helpful to see some reference to similar stats for other genes commonly studied for alternative splicing to give readers an idea of a "baseline" or at least what one should expect to see for a "random" gene. As someone not too familiar with the splicing literature, the variation in Figure 1A doesn't seem that small but this is likely because I don't know what numbers to expect. I'm also not sure about the relevance of the claim regarding "the likely presence of different levels of RBPs …" since I don't think it's clear which RBPs are involved in MAPT splicing (are these well characterized?, if yes, their levels can be quantified) and perhaps their levels are also fairly consistent across samples.

13. Results, second subsection: based on a finding that "66% of base pairs spanned less than 50 nucleotides and were contained within the exon units", the authors suggest that "the mature exons function as their own structural unit". However, looking at Figure 1B, there seem to be some high-probability, adjacent base-pairing interactions (green arcs, likely a stable helix) between exons 9 and 11 in isoform 3R, and similarly, 2 stable helices between exons 9 and 10 in isoform 4R. I recommend revising this part.

14. Results, third subsection: similarly to my previous comment, I also noticed strong base-pairing interactions (2 helices) between exon 10 and intron 10 in Figure 2. It would be helpful if the authors clarified this point and why they think that base pairing was "contained within exons, independent of introns."

Also, same question as above regarding the statement "despite likely differences in RBP concentrations, …".

15. Results, third subsection: it would be good to explicitly quantify (in the main text) the reactivity difference observed between the pre-mRNA and the mature isoform junctions, similarly to how this was done in a previous subsection for SSU vs. the isoforms, as it gives an idea of the degree of these changes and how strongly they suggest/support a higher pre-mRNA structure. That said, the statistical significance of such changes should be assessed/quantified in the context of the baseline (i.e., within-condition) variabilities.

16. Figure 2B: representative structures are shown for 3 clusters, but to me, it seems like there is a fourth cluster in the middle-right part of the plot (north east of the cluster whose representative structure is depicted at the bottom right). Was this cluster also identified by k-means? If yes, what is the representative structure and how relevant is it to the findings of this study? If this is not a separate cluster, it would help to clarify that the structure shown at the bottom right also corresponds to these points, or is it the structure at the top right corner? The latter option makes more sense based on the density plot (Figure 4C) for 4R mutant, but currently, no dashed line points directly to this region/cluster.

17. Figure 2C: the gel insets are mentioned briefly in the caption but I found it difficult to understand what these mean and why they are there. A bit more detail would be helpful, or otherwise perhaps move them to Supp. material and add some detail there.

18. Results, fourth subsection: in the statement about WT "The exon-intron junction of the representative structure for this region …" – does "this region" refer to the cloud in the middle? If yes, it's unclear from the previous sentence, which says WT had "structures distributed across the entire space…"

19. Results, fourth subsection: the last sentence refers to "the two other representative structures …" and Figure 2—figure supplement 2B, where the clustering results are shown more clearly and more than 3 clusters/structures emerge. This is really confusing, and I suggest re-thinking how to present the clustering and representative structures results. I find the supp figure to be clearer than Figure 2, even though more than 3 clusters are depicted. At the least, I think it could be helpful to clarify that more than 3 clusters were found and which structures represent which points/region in the plot.

20. Results, fifth subsection: the text on the non-synonymous and compensatory mutations isn't very clear. Were these all within exon 10? By compensatory mutations, do you mean double-mutants that recover the PSI?

21. Results, fifth subsection: did the author try to predict the PSI by using combinations of unfolding free energies (from several stages)?

22. Figure 3B: from the insets, it looks like the bootstrap variation estimates vary markedly differently for exonic vs. intronic mutations. Any idea why?

23. Figure 4A: what does "Experimental Label" mean?

24. Materials and methods, calculating \deltaG^{++} of unfolding of a region of interest: I found it difficult to follow the description of this calculation, so more detail would be appreciated. I'm also not sure what "base pairs within a region of interest were removed" means – do you mean the structure in that region was converted into a single-stranded RNA? Also, my understanding is that the notation \deltaG^{++} is for this particular non-equilibrium energy, however, I don't understand why the two numbers from which it is calculated are also denoted by \deltaG^{++}.

25. I have the impression that sometimes the authors omitted crucial explanations. For example, it is important to explicitly stated that inclusion of exon 10 results in the 4R isoform and that exclusion results in 3R early in the text to avoid confusions.

26 In Figure 1A, I suggest that splicing be drawn by having lines go from the 5' splice site of exon 9 to the 3' splice site on exon 10, or to that on exon 11, to show the alternative splicing forms.

27. Figure supplement 2A-B: space missing in "Invivo".

28. The manuscript could benefit from improvements to its writing to clarify and/or better explain a few points/statements and possibly also adjust some statements to better align with the analysis findings. Different subsections of the main text sometimes feel disconnected from the rest. Including the rationale of the authors in performing each analysis together with clear conclusions would go a long way helping the readers understand the various sections. The authors explain that the consistency of the splicing ratio (1:1) across tissues suggests that primary sequences and structure regulate this event, but not RNA binding proteins. It is strange that later on the authors include SRE (and binding of RNA binding proteins) as key regulators in their framework.

29. Line 51-52, the references for pre-mRNA structure should be Sun et al., 2019 NSMB and Liu et al., 2021, Genome Biology.

30. Shannon Entropy could be added along with BPP.

31. The PCR efficiency is normally associated with the size of fragment. The size of 3R is much shorter than that of 4R. Is there any estimations on the effects while comparing the structures of 3R and 4R?

32. Will the structure and SRE in the intron upstream of exon10 also affect the PSI?

---

## [Author Response]

Essential revisions:1. The figure of SSU structure could be improved with clear nucleotide information along with reactivities. The ROC shows that the sequence seems more important than DMS reactivities. It might be better to quantitatively measure how many single-stranded nucleotides have high DMS reactivities. The DMS reactivities for unpaired As and Cs are much higher than those for paired As and Cs. Is it due to protein protection? It seems unreasonable to compare the reactivity profiles between SSU and MARPT isoforms since the profile difference might be only due the different interacted proteins.

DMS selectively modifies unpaired A and Cs, so it is expected that unpaired A and Cs would have higher reactivity. Nonetheless, the reviewer brings up a good point about the effects of protein protection on DMS modification. To address this, we first extracted RNA from cells and removed the proteins, and then treated them with DMS (cell-free). We found that the increased DMS reactivities for unpaired ribonucleotides compared to paired was maintained in the cell-free condition (Figure 1 —figure supplement 7A) suggesting that the native structure of the RNA is contributing to the reactivity of these ribonucleotides rather than protein protection. This is consistent with previous in-cell vs. cell-free studies of RNAs.

We also modified the SSU structure in Figure 1 —figure supplement 6 to show clear nucleotide information with respective in-cell DMS reactivities and we thank the reviewer for this suggestion.

While we agree that the ROC indicates that sequence is marginally more accurate in predicting the SSU structure compared to just using DMS reactivities, the goal of this analysis was to show that combining the sequence and DMS reactivities yielded the best prediction accuracy of the SSU structure. This is further supported by the ROC generated using cell-free SSU DMS reactivities, consistent with previously published works that show combining experimental and computational models yields the most accurate predictions (Hajdin et al., 2013; Sükösd et al., 2013).

This new data is now included in the manuscript on page 6, lines 151-167.

“As an internal control for our probing experiments, we also collected DMS-MaP data for the small subunit ribosomal RNA (SSU) (Figure 1—figure supplement 6), which has a

well-defined secondary structure (Petrov et al., 2014). As expected, the DMS reactivities of unpaired nucleotides were significantly higher than for paired nucleotides both for RNA probed in cells and for RNA isolated from cells prior to probing (Figure 1–figure supplement 7A). This experiment confirmed that our DMS probing recapitulates native RNA secondary structure regardless of the presence of proteins, consistent with Previous studies (Woods et al., 2017; Lackey et al., 2018). We used the SSU in-cell reactivity data to calibrate the estimation of equilibrium ensembles (Materials and methods), and we confirmed that structure modeling guided by experimental DMS reactivities yielded a more accurate estimation of the SSU structure than the model not informed by chemical probing data (Figure 1—figure supplement 7B).

The median in-cell DMS reactivity of the mature MAPT isoforms was 0.22, significantly greater than the median in-cell DMS reactivity of the SSU, which was 0.008 (Figure 1– figure supplement 7C). This difference was recapitulated in cell-free samples (Figure 1– figure supplement 7C).”

2. It is interesting that 66% of base pairs were contained within exon units. Would the percentage remain similar when performing RNA folding with sliding window? If so, this observation could be explained as co-transcriptional folding.

This is an excellent question that we agree could benefit from further analysis. To model co-transcriptional folding, we first folded Exon 10 and counted the number of base pairs. We then “in silico” transcribed Exon 11 one nucleotide at a time and computed the number of intra-exon base pairs (base pairs just within Exon 10 or base pairs just within Exon 11) for every additional nucleotide. We plotted the percentage of intra-exon base pairs out of the total number of base pairs found for the sequence at each nucleotide of Exon 11 added (Figure 1—figure supplement 8). We found that while the total number of intra-exon base pairs decreased as Exon 11 is transcribed, once Exon 11 is complete 88.9% of all base within exon units once the full length mRNA is folded.

We also in silico transcribed the pre-cursor mRNA co-transcriptionally by adding Intron 10 one nucleotide at a time to Exon 10 and then calculated the number of intraexon/ intron base pairs as a percentage of the total number of base pairs (Figure 2 – figure supplement 2). There is a large drop in intra exon/intron base pairs 37 nucleotides into intron 10 but ultimately 65.7% of all base pairs are self-contained within an exon or intron.

It is important to note that the 66% intra-exon pairs initially reported was averaged over all exons probed in this manuscript, we now provide these measurements for specific sequences.

We have modified the text and figure legends accordingly. For example, on page 6 lines 170-174:

“In the 4R isoform, approximately 89% of base pairs were contained within the exon units; only 11% of base pairs were between residues from exon 10 with those of exon 11 (Figure 1—figure supplement 8). This result suggests that the mature exons fold as independent structural units.”

3. The intron is more structured than the exon. Rather than RNA structure, is it possible that it is due to the protein protection? Would it remain a similar pattern under deproteinized conditions?

This is an excellent point, and to address this we now include cell-free analysis of exon 10 – intron 10 junction. Interestingly, we found the cell-free and in-cell DMS reactivities to be highly correlated (Figure 2 —figure supplement 1D). Furthermore, DMS reactivities in the pre-cursor MAPT transcript were lower than the mature MAPT transcripts implying that the pre-cursor mRNA is more structured than the mature mRNA both in-cell and cell-free (Figure 2 —figure supplement 3). Thus, protein protection does not appear to have a large effect on these RNA’s DMS reactivity. This is now discussed on page 9, lines 243-248.

“However, when we compared DMS reactivities for pre-mRNA and the mature 4R isoform, we found that DMS reactivity in exon 10 was significantly lower for the pre-mRNA (median in-cell DMS reactivity: 0.08) than for the 4R isoform (median in-cell DMS reactivity: 0.22) (Figure 2—figure supplement 3). This was also the case for RNA probed under cell-free conditions (Figure 2—figure supplement 1D). The pre-mRNA is thus apparently more structured than mature mRNA independent of protein protection.”

4. The unfolding free energy is based on the first step of 5'SS recognition (9 nts) for U1 snRNA. It is also possible that the RNA structure only affects U5 snRNA binding? The authors could test unfolding 3nt and see whether it will be similar or no effect. Again, will the clusters remain similar if the folding windows change?

The t-SNE dimensionality reduction and subsequent k-means clustering was performed on the 201-nucleotide region flanking exon 10 – intron 10 junction. No max pairing distance was used to fold the sequence. Each point on the plot represents one of the 1000 suboptimal structures sampled and the point was colored based on the unfolding free energy of the 5’SS (9 nucleotides). Hence, unfolding a different region other than the 5’ SS would change the color of each point but it would not affect the position of the points or the clusters. We think it may get confusing to the reader if we provide in the supplement differently colored versions of this plot based on unfolding energy. However, we do address these questions in Figure 3.

For our model to predict PSI, we calculated unfolding free energies of regions around the exon-intron junction through different stages of the splicing cycle (Figure 3A). We use cryo-EM structures of these stages to identify unpaired nucleotides in the spliceosome. U5 snRNA arrives in the splicing cycle during the assembly of the Pre-B spliceosomal complex when the 5’SS is transferred from the U1snRNA to the U6 and U5 snRNA (Bertram et al., 2017). Hence, the RNA’s interaction with the U5 snRNA would be captured by the nucleotides unpaired in the pre-B complex. Furthermore, our analysis finds that the model with the best prediction accuracy is when unfolding a much larger region of the exon-intron junction found in the Bact complex (43 nucleotides, 10 exonic, 33 intronic). This suggests that structures distal to the 5’ splice site are impacting Exon 10 splicing (Figure 3B, C), and we discuss this on page 32 of the main text.

5. It is not necessary to unfold all the nucleotides at one time. At the different steps during the splicing, the RNA structure might be very dynamic and remodelled in the process.

We agree with the reviewer that all nucleotides are probably not accessible at all times and that RNA structure is dynamic. However, we needed a systematic approach to identify which nucleotides should be unfolded for the structure model during the splicing cycle. Hence, we performed an analysis of Cryo-EM structures of the human spliceosome at various stages of the splicing model to identify the region of the precursor mRNA with no intramolecular contacts at each stage of the splicing cycle (Figure 3A). The best predictive performance observed when we unfolded all 43 nucleotides found associated with the Bact complex (Figure 3B, C) suggesting that the pre-mRNA needs to be unpaired to allow the spliceosome to rearrange the conformation. We now discuss this on page 32, lines 775-781:

“Our finding corroborates evidence that RNA structure near this exon-intron junction is extensive (Tan et al., 2019). Note that we do not claim that all 43 nucleotides need to remain fully unpaired during the splicing cycle, as the entire cycle is dynamic and likely involves other intermediate structures. Rather, our model argues that mRNA unfolding and accommodation into the Bact complex is a key rate limiting step in splicing, and considering this step is necessary to accurately model splicing outcome for a diverse set of mutations.”

6. For the training set, is there any reason for using 20 synonymous and intronic mutations? A detailed clarification is necessary. Why not non-synonymous mutations?

Non-synonymous variants are primarily implicated in altering the amino acid sequence of protein translated from the region. Previous studies have shown that synonymous and intronic variants are more likely to affect splicing (Supek et al., 2014; H. Lin et al., 2019a). Additionally, these variants are causative of disease through structural changes as shown by a recent analysis of synonymous cancer mutations (Sharma et al., 2019) as well as an investigation into pathogenic intronic variants (C. L. Lin, Taggart, and Fairbrother 2016). Hence, in order to train a model that only uses structural information and select the region among the four spliceosomal complexes that would best predict MAPT Exon 10 splicing, we used only 20 synonymous and intronic variants to eliminate the possibility of the pathogenicity arising from other effects such as changes in the protein sequence. Ultimately, we do validate our structure model on non-synonymous variants (Figure 4 —figure supplement 1A, B).

We have modified the text accordingly. On page 15, lines 370-375:

“To identify the spliceosome complex footprint that best predicts splicing outcome, we examined the relationship between unfolding energy and splicing outcome for 20 synonymous or intronic mutations in exon 10 and intron 10 (Figure 3—figure supplement 1A). These mutations are more likely to affect splicing (Supek et al., 2014; H. Lin et al., 2019b) and structure (Sharma et al., 2019; C. L. Lin, Taggart, and Fairbrother 2016) than mutations that alter the protein sequence.”

7. The prediction could be improved by taking into account RBPs. The different DMS reactivity profiles are likely due to RBP binding. See the comment above.

We would like to direct the reviewer to Figure 4 —figure supplement 2 and lines 631-641 in the main text. We used motif strengths of RBPs previously identified as regulating splicing of the MAPT Exon 10 – Intron 10 junction in our β regression framework. This unfortunately resulted in poor prediction accuracy of MAPT Exon 10 PSI with a correlation coefficient of 0.08, hence our use of splicing regulatory elements.

We now address the issue of DMS reactivity differences due to RBP binding by analyzing the DMS reactivity profiles of the MAPT mature and pre-cursor transcripts in deproteinized conditions (cell-free) as described above in reviewer comment #3 (Figure 1 —figure supplement 7C, Figure 2 —figure supplement 3). We find that DMS reactivity is not specifically affected by RBP binding within cells under our conditions.

8. Figure 6E. The first mutation the model predicted as “No change” seems to change; the splicing ratio does change at the leftmost column, although to a lesser extent than in the other columns called as “changed”.

We agree with the reviewer that the mutation +23U>C predicted as “no change” is lower than the WT splicing ratio. However, when we performed a two-tailed Wilcoxon-Rank Sum test over three replicates, we obtained a p-value of 1 for +23U>C, suggesting that the decrease in ratio is not significant. This was also the case for +26G>A. Hence, we concluded that we had confirmed our prediction that these two mutations resulted in no change to the Exon 10 splicing ratio. We have now added the p-value of 1 for those two “No change” mutations on Figure 6 —figure supplement 1 and modified the figure legend.

9. I would encourage the authors to come up with an explanation/hypothesis of why some mutations are more impactful, with respect to disease, than others.

This is a good point and one we have struggled with during the drafting of the manuscript. It is quite clear the severity of disease is not always correlated to the 3R/4R ratio. Our data of “healthy” GTeX individuals (Figure 1A) shows variation in this ratio among these healthy individuals. We have added the following text to the discussion in response to this comment on page 34, line 825-833 of the discussion in the main text:

“Thus, although our model combines both structural and sequence features to achieve quantitative prediction accuracy of the 3R to 4R ratio for a wide range of disease mutations (synonymous, non-synonymous, intronic and exonic), it is not clear that PSI alone is predictive of severity of disease for the broad class of tauopathies (Majounie et al. 2013). Disease severity is compounded by other factors including gene-gene interactions and environmental factors. As such, the value of our model stems more from how it incorporates RNA structure in predicting alternative splicing, rather than as a direct predictor of disease severity.”

10. Throughout the text, the authors discuss RNA conformations as being “more” or “less” structured and then “structured” or “unstructured.” It would be beneficial to describe what it means to be “structured” versus “non-structured” and if they are truly the extremes or another way of saying more of less structured. It is difficult to follow as the text flips between these four ways to describe the structure, and then incorporates the direct measurement of the energy of splice site unfolding.

The reviewer makes an excellent point. We were primarily using unstructured versus structured when describing the interpretation of high and low DMS reactivities. To the reviewer’s point, DMS reactivities are more probabilistic in nature and hence we have gone through the text and removed uses of “unstructured” and replaced them with “less structured.”

For example, in the figure caption of Figure 2A, page 13, lines 295-298:

“Each value is shown with its standard error and colored by reactivity based on the color scale. High median DMS reactivities correspond to less structured regions, whereas low median DMS reactivities correspond to more structured regions.”

11. It would be good to include “MAPT” in the title considering all data was collected for this gene. If the author wanted to present this framework as a general model for splicing prediction, analysis of a second exon-intron junction would be required.

This is a good point. We have changed the title of the manuscript to: “Quantitative prediction of variant effects on alternative splicing in MAPT using endogenous pre messenger RNA structure probing”.

Given the current length and complexity of the manuscript we think extending our

analysis to other systems falls outside the scope of the manuscript.

12. Results, first subsection: MAPT’s PSI variability was analyzed in the context of different individuals and different cell types/tissues. This is followed by a statement that this isoform ratio is consistent. It would be helpful to see some reference to similar stats for other genes commonly studied for alternative splicing to give readers an idea of a “baseline” or at least what one should expect to see for a “random” gene. As someone not too familiar with the splicing literature, the variation in Figure 1A doesn’t seem that small but this is likely because I don’t know what numbers to expect. I’m also not sure about the relevance of the claim regarding “the likely presence of different levels of RBPs …” since I don’t think it’s clear which RBPs are involved in MAPT splicing (are these well characterized?, if yes, their levels can be quantified) and perhaps their levels are also fairly consistent across samples.

We thank the reviewer for this excellent point. Mele et. Al concluded after using linear mixed models to analyze the contribution of tissue and individual differences in transcript isoform variation of GteX RNA-Seq data (Melé et al., 2015) that splice isoform expression has little variation among both individuals and tissues. However, they did not provide a baseline number. To address this point, we calculated and plotted the distribution of PSI of two other exons in MAPT across 15 tissues in Figure 1 – figure supplement 2. Exon 4 is constitutively spliced and always included in every transcript. Exon 2 is alternatively spliced. Our analysis showed that Exon 4 PSI had low variability across different brain tissues with an overall standard deviation of 0.1 while Exon 2 had higher variability with a standard deviation of 0.3. We found that MAPT Exon 10 falls in the middle with a standard deviation of 0.2.

We have modified the text and figure legends accordingly. For example, on page 5 lines 129-131:

“Furthermore, exon 10 inclusion variability (0.2) was between the variability for a MAPT constitutively spliced exon (0.1) and another MAPT alternatively spliced exon (0.3) (Figure 1—figure supplement 2).”

Furthermore, we analyzed the overall expression levels of 9 RBPs previously implicated in regulating alternative splicing of MAPT Exon 10 from RNA-Seq data in the GTeX database. As can be seen from the violin plots (Figure 1 —figure supplement 3), there is variability of RBP expression levels among different tissues and individuals. We did attempt to model Exon 10 PSI unsuccessfully using RBP expression levels initially during the development of the manuscript but were unsuccessful in finding any meaningful correlations. This indicated to us that RBP levels alone are not a good predictor of PSI in this case. However, given the complexity and length of the manuscript we did not include details of this negative result in the manuscript.

We have modified the text and figure legends accordingly. For example, on page 5 lines 131-133:

“As levels of RBP expression varied considerably across individuals and tissues (Figure 1—figure supplement 3), sequence and structural features of the MAPT pre-mRNA also likely regulate inclusion of exon 10.”

13. Results, second subsection: based on a finding that "66% of base pairs spanned less than 50 nucleotides and were contained within the exon units", the authors suggest that "the mature exons function as their own structural unit". However, looking at Figure 1B, there seem to be some high-probability, adjacent base-pairing interactions (green arcs, likely a stable helix) between exons 9 and 11 in isoform 3R, and similarly, 2 stable helices between exons 9 and 10 in isoform 4R. I recommend revising this part.

We agree that while there are a few high probability base pairs between exon 9 and exon 11 as well as exon 9 and exon 10, overall, we found a larger number of intra-exon base pairs compared to inter-exon base pairs. As described in response to reviewer comment #2, we performed co-transcriptional folding to better characterize the intra vs inter exon base-pairs.

14. Results, third subsection: similarly to my previous comment, I also noticed strong base-pairing interactions (2 helices) between exon 10 and intron 10 in Figure 2. It would be helpful if the authors clarified this point and why they think that base pairing was "contained within exons, independent of introns."Also, same question as above regarding the statement "despite likely differences in RBP concentrations, …".

Please refer to reviewer comment #2, for how we performed additional analysis to show base pairing is contained within exons and introns. Please refer to reviewer comment #12 for how we showed that there are differences in RBP concentrations between tissues and individuals.

15. Results, third subsection: it would be good to explicitly quantify (in the main text) the reactivity difference observed between the pre-mRNA and the mature isoform junctions, similarly to how this was done in a previous subsection for SSU vs. the isoforms, as it gives an idea of the degree of these changes and how strongly they suggest/support a higher pre-mRNA structure. That said, the statistical significance of such changes should be assessed/quantified in the context of the baseline (i.e., within-condition) variabilities.

We thank the reviewer for the pointing this out to us and we have modified the text as

follows. On page 9, lines 243-246:

“However, when we compared DMS reactivities for pre-mRNA and the mature 4R isoform, we found that DMS reactivity in exon 10 was significantly lower for the pre-mRNA (median in-cell DMS reactivity: 0.08) than for the 4R isoform (median in-cell DMS reactivity: 0.22) (Figure 2—figure supplement 3).”

16. Figure 2B: representative structures are shown for 3 clusters, but to me, it seems like there is a fourth cluster in the middle-right part of the plot (north east of the cluster whose representative structure is depicted at the bottom right). Was this cluster also identified by k-means? If yes, what is the representative structure and how relevant is it to the findings of this study? If this is not a separate cluster, it would help to clarify that the structure shown at the bottom right also corresponds to these points, or is it the structure at the top right corner? The latter option makes more sense based on the density plot (Figure 4C) for 4R mutant, but currently, no dashed line points directly to this region/cluster.

We have altered Figure 2B to show five clusters very clearly by circling them. Furthermore, we took representative structures in Figure 2B and moved them down to Figure 2D and included all 5 centroid structures.

For example, on page 10 lines 267-269:

“We visualized the structural ensemble for the 3000 structures using t-Distributed stochastic neighbor embedding (t-SNE) (Van Der Maaten and Hinton 2008) and identified five clusters (Figure 2B; Materials and methods).”

17. Figure 2C: the gel insets are mentioned briefly in the caption but I found it difficult to understand what these mean and why they are there. A bit more detail would be helpful, or otherwise perhaps move them to Supp. material and add some detail there.

We added the gels to make it clear that these specific mutations +19A>C and +15G>C were indeed affecting the expression of the two isoforms compared to the wild type sequence. We added additional detail in the figure captions. On page 13-14, lines 314- 322:

“Inserts are gel images from representative of splicing assays using a reporter plasmid expressing either the wild-type sequence (WT), the +19C>G (3R) mutation or +15A>C (4R) mutation in HEK293 cells, where the RNA was extracted and reverse transcribed to measure the isoform ratio using specific PCR amplification (Materials and methods). In WT, both 3R (Exon 9 – Exon 11) and 4R (Exon 9 – Exon 10 – Exon 11) isoforms are expressed (two bands). In the presence of the 3R mutation, only the 3R isoform is expressed (one band) whereas for the 4R mutation only the 4R isoform is expressed (one band). Gel insets for the 3R and 4R mutation are in their respective density plots.”

18. Results, fourth subsection: in the statement about WT "The exon-intron junction of the representative structure for this region …" – does "this region" refer to the cloud in the middle? If yes, it's unclear from the previous sentence, which says WT had "structures distributed across the entire space…"

We thank the reviewer for pointing this out and upon reading this paragraph again, we realize the wording was not clear. We added additional labels to Figure 2B to clearly point out the different clusters and we now make use of those labels to refer to the clusters in our main text.

For example, on page 10-11, lines 277-286:

“The wild-type sequence forms structures distributed across the entire space with about 70% of structures found in Clusters 2, 3, and 4 (Figure 2—figure supplement 4B). By contrast, in the +19C>G mutant that strongly favors the 3R isoform (Tan et al., 2019), more than 55% of structures belong to Cluster 1, which is defined by a fully base-paired 5’ splice site (Figure 2D). Conversely, greater than 50% of structures in the ensemble of the +15 A>C mutant (Cluster 5), which shifts the isoform balance entirely to 4R (Tan et al. 2019), were characterized by lower DG‡ of unfolding for the splice site region (Figure 2B, C). Correspondingly, the 5’ splice site for the Cluster 5 representative structure was less structured than that of Cluster 1 (Figure 2D).”

19. Results, fourth subsection: the last sentence refers to "the two other representative structures …" and Figure 2—figure supplement 2B, where the clustering results are shown more clearly and more than 3 clusters/structures emerge. This is really confusing, and I suggest re-thinking how to present the clustering and representative structures results. I find the supp figure to be clearer than Figure 2, even though more than 3 clusters are depicted. At the least, I think it could be helpful to clarify that more than 3 clusters were found and which structures represent which points/region in the plot.

We agree with the reviewer that this figure and the corresponding paragraph in the main text is unclear. It makes sense to show all five clusters and structures like the supplementary figure. We have thus made changes as described in reviewer comments #16-#18 to Figure 2.

20. Results, fifth subsection: the text on the non-synonymous and compensatory mutations isn't very clear. Were these all within exon 10? By compensatory mutations, do you mean double-mutants that recover the PSI?

The non-synonymous mutations were within exon 10 while the compensatory mutations which are double mutations intended to rescue Exon 10 PSI changes, spanned across both exon 10 and intron 10. We have added the following to the main text to make this clearer. On page 16, lines 395-398:

“We then tested the structural ensemble-based model on an additional 24 non-synonymous and compensatory mutations found in exon 10 and intron 10.

Compensatory mutations are double mutations that were designed to rescue changes in exon 10 splicing caused by a single mutation (Grover et al., 1999).”

21. Results, fifth subsection: did the author try to predict the PSI by using combinations of unfolding free energies (from several stages)?

While this is an interesting idea, we did not do this because the unfolded region at each subsequent stage of splicing is correlated with the previous stages. Hence, the structural features would not be independent of each other when training the model which can affect the β regression framework.

22. Figure 3B: from the insets, it looks like the bootstrap variation estimates vary markedly differently for exonic vs. intronic mutations. Any idea why?

The intronic mutations are located closer to the exon-intron junction (mean distance from EIJ: 14) while the exonic synonymous mutations are further from the junction (mean distance from EIJ: 54), and this is driven by one mutation in particular that is 64 nucleotides upstream of the junction. We see that the correlation coefficient between experimental and predicted Exon 10 PSIs for the synonymous mutations increases as we unfold larger regions around the exon-intron junction (Figure 3B) and the variation decreases suggesting that the synonymous mutations are affecting distal structures.

We added the following sentence in the main text to explain this. On page 16, lines 389-393:

“Synonymous mutations that alter exon 10 inclusion lie a mean distance of 54 nucleotides from the exon-intron junction, whereas those in the intron are a mean of 14 nucleotides from the junction. The variation in bootstrapped correlation coefficients decreased as a larger region around the exon-intron junction was unfolded, suggesting that the synonymous mutations affect distal structures.”

23. Figure 4A: what does “Experimental Label” mean?

We searched the literature to find the 47 mutations around the MAPT Exon 10 Intron 10 junction that had been experimentally validated using splicing assays. The experimental label is what each study had labelled and called the mutation: a 3R mutation shifted the isoform balance significantly towards the 3R isoform, a 4R mutation shifted the isoform balance towards the 4R isoform while a 50-50 mutation maintained the isoform balance.

We have added text in the figure caption for Figure 4A to explain Experimental Label as follows on page 22 lines 535-538:

“Mutation Type refers to whether the mutation is exonic non-synonymous, exonic synonymous, intronic or compensatory. Experimental Label is the label given by the original study that experimentally validated each mutation using a splicing assay.”

24. Materials and methods, calculating \deltaG^{++} of unfolding of a region of interest: I found it difficult to follow the description of this calculation, so more detail would be appreciated. I’m also not sure what “base pairs within a region of interest were removed” means – do you mean the structure in that region was converted into a single-stranded RNA? Also, my understanding is that the notation \deltaG^{++} is for this particular non-equilibrium energy, however, I don’t understand why the two numbers from which it is calculated are also denoted by \deltaG^{++}.

We agree with the reviewer that section could have been written more clearly. We have modified this section as follows on page 43, lines 1091-1103:

The DG‡ of unfolding energies of regions of interest were calculated using a custom Python script. The non-equilibrium unfolding energy of the region, defined as the energy require to unfold a specific region without allowing refolding (Mustoe et al., 2018) is defined as follows:

ΔG‡ = ΔGfold − ΔDGunfold

The ΔG of the original folded structure (ΔG^fold^) was calculated with the efn2 program in RNAstructure (Reuter and Mathews 2010). Next, the base pairs within a region of interest were made single stranded by setting the base pair column value to be 0 in the CT file. From this modified CT file, we evaluate the DG of the unfolded structure (DG^unfold^) with efn2. This was done for every suboptimal structure in the Boltzmann ensemble. For example, to determine the ΔG‡ of unfolding of the splice site, we made all nucleotides within the last 3 nucleotides of the exon and the first 6 nucleotides of the intron single stranded.

25. I have the impression that sometimes the authors omitted crucial explanations. For example, it is important to explicitly stated that inclusion of exon 10 results in the 4R isoform and that exclusion results in 3R early in the text to avoid confusions.

The reviewer brings up a good point that we were not clear in our introduction of 3R vs 4R MAPT isoforms.

We have addressed this on page 3, lines 82-85:

“Exons 9, 10, 11, and 12 encode the critical microtubule binding repeat domain in Tau. Exons 9, 11, and 12 are constitutively spliced, but exon 10 is alternatively spliced resulting in MAPT isoforms with either four microtubule binding repeats (4R) or three repeats (3R) when exon 10 is included or skipped, respectively.”

26 In Figure 1A, I suggest that splicing be drawn by having lines go from the 5' splice site of exon 9 to the 3' splice site on exon 10, or to that on exon 11, to show the alternative splicing forms.

We thank the reviewer for the feedback and have modified Figure 1A to show the two alternative splicing isoforms. We also modified the figure caption as follows, on page 7, lines 183-186:

“Exon 10 is alternatively spliced to form the 3 repeat (3R) or 4 repeat (4R) isoform. This is highlighted by the alternate lines from the 5’ splice site of Exon 9 to either the 3’ splice site of Exon 10 (4R) or the 3’ splice site of Exon 11 (3R).”

27. Figure supplement 2A-B: space missing in "Invivo".

We thank the reviewer for pointing this out and have made the modifications on Figure 1—figure supplement 4A and B as well as Figure 2 —figure supplement 1A and B. We have additionally changed ‘in vivo’ to in-cell and ‘ex vivo’ to cell-free throughout the text to accurately capture the experiments performed and avoid confusion.

28. The manuscript could benefit from improvements to its writing to clarify and/or better explain a few points/statements and possibly also adjust some statements to better align with the analysis findings. Different subsections of the main text sometimes feel disconnected from the rest. Including the rationale of the authors in performing each analysis together with clear conclusions would go a long way helping the readers understand the various sections. The authors explain that the consistency of the splicing ratio (1:1) across tissues suggests that primary sequences and structure regulate this event, but not RNA binding proteins. It is strange that later on the authors include SRE (and binding of RNA binding proteins) as key regulators in their framework.

We employed the services of a copy editor to edit our text which has greatly streamlined it. Moreover, based on reviewer comment #12, we analyzed the expression of RBPs implicated in MAPT Exon 10 splicing regulation and found them to be different between tissues and individuals (Figure 1 —figure supplement 3) and RBP expression levels alone could not be used to predict MAPT Exon 10 PSI. Hence, we used the changes induced by mutations to the motif strength of generalized splicing regulatory enhancers and silencers to predict Exon 10 splicing changes.

We modified our text as follows on page 20, lines 468-476 as follows:

“Exon 10 splicing is highly regulated by differential binding of RBPs to cis-SREs within exon 10 and intron 10 (Qian and Liu 2014). The expression patterns of RBPs known to bind MAPT pre-mRNA vary across tissues and individuals (Figure 1—figure supplement 3) and are not predictive of exon 10 PSI. Additionally, while our structure-only model performs moderately well for 47 mutations (R2=0.74) (Figure 4—figure supplement 1A, see Supplementary file 2 for further details about mutations), the structure only model performs particularly poorly for non-synonymous mutations (median bootstrapped R2 = - 0.21, Figure 4—figure supplement 1A). Hence, we hypothesized that consideration of mutation-induced changes in binding of SREs might improve our model.”

29. Line 51-52, the references for pre-mRNA structure should be Sun et al., 2019 NSMB and Liu et al., 2021, Genome Biology.

We have added the references on page 2, lines 54-56:

“Only recently has high-resolution in-cell experimental characterization been applied to pre-mRNA structure determination (Mustoe et al., 2018; Sun et al., 2019; Liu et al., 2021; Bubenik et al., 2020).”

30. Shannon Entropy could be added along with BPP.

Shannon entropy is generally computed over a 50 nucleotide (nt) window. Most of the structures we analyzed in this manuscript are around 200 nt in length, so providing windowed Shannon entropy for these shorter amplicons really does not add much information, e.g. the Shannon entropy is low for the entire exon 10 – intron 10 junction.

31. The PCR efficiency is normally associated with the size of fragment. The size of 3R is much shorter than that of 4R. Is there any estimations on the effects while comparing the structures of 3R and 4R?

Previous studies have shown that PCR efficiency does not affect the mutation rates measured by chemical probing of these fragments (Smola et al., 2015). Moreover, we used a low number of PCR cycles (25) to amplify the two regions prior to amplicon sequencing.

32. Will the structure and SRE in the intron upstream of exon10 also affect the PSI?

The reviewer makes an excellent point; the structure and SRE around the 3’ splice site the intron 9 – exon 10 junction would affect the MAPT Exon 10 PSI. However, given the length and complexity of the current manuscript, this additional analysis falls outside the scope. Furthermore, the 3’ splice site is more complex since it includes both the branchpoint sequence and the polypyrimidine tract and the exact location of these important sequences are not well defined for this junction.

We added the following text to the discussion to address this. On page 33-34, lines 815-

818:

“Another limitation is that the current model does not consider structural and sequence features around the 3’ splice site (in the case of MAPT exon 10, the intron 9 – exon 10 junction), that are expected to impact exon 10 splicing regulation.”

References

Bertram, Karl, Dmitry E. Agafonov, Olexandr Dybkov, David Haselbach, Majety N. Leelaram, Cindy L. Will, Henning Urlaub, Berthold Kastner, Reinhard Lührmann, and Holger Stark. 2017. “Cryo-EM Structure of a Pre-Catalytic Human Spliceosome Primed for Activation.” *Cell* 170 (4): 701-713.e11. https://doi.org/10.1016/j.cell.2017.07.011.

Bubenik, Jodi L., Melissa Hale, Ona Mcconnell, Eric T. Wang, Maurice S. Swanson, Robert C. Spitale, and J. Andrew Berglund. 2020. “RNA Structure Probing to Characterize RNA-Protein Interactions on a Low Abundance Pre-MRNA in Living Cells.” *RNA (New York, N.Y.)* 27 (3): 343–58. https://doi.org/10.1261/RNA.077263.120.

Grover, Andrew, Henry Houlden, Matt Baker, Jennifer Adamson, Jada Lewis, Guy Prihar, Stuart Pickering-Brown, Karen Duff, and Mike Hutton. 1999. “5’ Splice Site

Mutations in Tau Associated with the Inherited Dementia FTDP-17 Affect a StemLoop Structure That Regulates Alternative Splicing of Exon 10*.” *Journal of Biological Chemistry* 274 (21): 15134–43. http://www.jbc.org/content/274/21/15134.full.pdf.

Hajdin, Christine E, Stanislav Bellaousov, Wayne Huggins, Christopher W Leonard, David H Mathews, and Kevin M Weeks. 2013. “Accurate SHAPE-Directed RNA Secondary Structure Modeling, Including Pseudoknots.” *Proceedings of the National Academy of Sciences of the United States of America* 110 (14): 5498– 5503. https://doi.org/10.1073/pnas.1219988110.

Lackey, Lela, Aaztli Coria, Chanin Woods, Evonne McArthur, and Alain Laederach. 2018. “Allele-Specific SHAPE-MaP Assessment of the Effects of Somatic Variation and Protein Binding on MRNA Structure.” *RNA (New York, N.Y.)* 24 (4): 513–28. https://doi.org/10.1261/rna.064469.117.

Lin, Chien Ling, Allison J Taggart, and William G Fairbrother. 2016. “RNA Structure in Splicing: An Evolutionary Perspective.” *RNA Biology* 13 (9): 766–71. https://doi.org/10.1080/15476286.2016.1208893.

Lin, Hai, Katherine A. Hargreaves, Rudong Li, Jill L. Reiter, Yue Wang, Matthew Mort, David N. Cooper, et al. 2019a. “RegSNPs-Intron: A Computational Framework for Predicting Pathogenic Impact of Intronic Single Nucleotide Variants.” *Genome Biology* 20 (1): 1–16. https://doi.org/10.1186/S13059-019-1847-4/FIGURES/5.

Lin, Hai, Katherine A Hargreaves, Rudong Li, Jill L Reiter, Yue Wang, Matthew Mort, David N Cooper, et al. 2019b. “RegSNPs-Intron: A Computational Framework for Predicting Pathogenic Impact of Intronic Single Nucleotide Variants.” *Genome Biology* 20 (1). https://doi.org/10.1186/s13059-019-1847-4.

Liu, Zhenshan, Qi Liu, Xiaofei Yang, Yueying Zhang, Matthew Norris, Xiaoxi Chen,

Jitender Cheema, Huakun Zhang, and Yiliang Ding. 2021. “In Vivo Nuclear RNA

Structurome Reveals RNA-Structure Regulation of MRNA Processing in Plants.” *Genome Biology* 22 (1): 1–22. https://doi.org/10.1186/S13059-020-022364/FIGURES/5.

Maaten, Laurens Van Der, and Geoffrey Hinton. 2008. “Visualizing Data Using T-SNE.” *Journal of Machine Learning Research* 9: 2579–2605. https://lvdmaaten.github.io/publications/papers/JMLR_2008.pdf.

Majounie, Elisa, William Cross, Victoria Newsway, Allissa Dillman, Jana Vandrovcova, Christopher M. Morris, Michael A. Nalls, et al. 2013. “Variation in Tau Isoform Expression in Different Brain Regions and Disease States.” *Neurobiology of Aging* 34 (7): 1922.e7-1922.e12. https://doi.org/10.1016/j.neurobiolaging.2013.01.017.

Melé, Marta, Pedro G Ferreira, Ferran Reverter, David S Deluca, Jean Monlong, Michael Sammeth, Taylor R Young, et al. 2015. “The Human Transcriptome across Tissues and Individuals.” *Science* 348 (6235): 660–665. https://doi.org/10.1126/science.aaa0355.

Mustoe, Anthony M., Steven Busan, Greggory M. Rice, Christine E. Hajdin, Brant K. Peterson, Vera M. Ruda, Neil Kubica, Razvan Nutiu, Jeremy L. Baryza, and Kevin M. Weeks. 2018. “Pervasive Regulatory Functions of MRNA Structure Revealed by High-Resolution SHAPE Probing.” *Cell* 173 (1): 181-195.e18. https://doi.org/10.1016/j.cell.2018.02.034.

Petrov, Anton S., Chad R. Bernier, Burak Gulen, Chris C. Waterbury, Eli Hershkovits, Chiaolong Hsiao, Stephen C. Harvey, et al. 2014. “Secondary Structures of RRNAs from All Three Domains of Life.” *PLoS ONE* 9 (2): e88222. https://doi.org/10.1371/journal.pone.0088222.

Qian, Wei, and Fei Liu. 2014. “Regulation of Alternative Splicing of Tau Exon 10.” *Neuroscience Bulletin* 30 (2): 367–77. https://doi.org/10.1007/s12264-013-1411-2.

Reuter, Jessica S, and David H Mathews. 2010. “RNAstructure: Software for RNA Secondary Structure Prediction and Analysis.” *BMC Bioinformatics* 11 (1): 1–9. https://doi.org/10.1186/1471-2105-11-129.

Sharma, Yogita, Milad Miladi, Sandeep Dukare, Karine Boulay, Maiwen CaudronHerger, Matthias Groß, Rolf Backofen, and Sven Diederichs. 2019. “A Pan-Cancer Analysis of Synonymous Mutations.” *Nature Communications* 10 (1): 1–14. https://doi.org/10.1038/s41467-019-10489-2.

Smola, Matthew J, Greggory M Rice, Steven Busan, Nathan A Siegfried, and Kevin M Weeks. 2015. “Selective 2′-Hydroxyl Acylation Analyzed by Primer Extension and Mutational Profiling (SHAPE-MaP) for Direct, Versatile and Accurate RNA Structure Analysis.” *Nature Protocols* 10 (11): 1643–69. https://doi.org/10.1038/nprot.2015.103.

Sükösd, Zsuzsanna, M Shel Swenson, Jørgen Kjems, and Christine E Heitsch. 2013. “Evaluating the Accuracy of SHAPE-Directed RNA Secondary Structure Predictions.” *Nucleic Acids Research* 41 (5): 2807–16. https://doi.org/10.1093/nar/gks1283.

Sun, Lei, Furqan M. Fazal, Pan Li, James P. Broughton, Byron Lee, Lei Tang, Wenze Huang, Eric T. Kool, Howard Y. Chang, and Qiangfeng Cliff Zhang. 2019. “RNA Structure Maps across Mammalian Cellular Compartments.” *Nature Structural and Molecular Biology* 26 (4): 322–30. https://doi.org/10.1038/s41594-019-0200-7.

Supek, Fran, Belén Miñana, Juan Valcárcel, Toni Gabaldón, and Ben Lehner. 2014. “Synonymous Mutations Frequently Act as Driver Mutations in Human Cancers.” *Cell* 156 (6): 1324–35. https://doi.org/10.1016/J.CELL.2014.01.051/ATTACHMENT/3FCA62C2-C60145EF-A9A7-821D41838468/MMC3.XLSX.

Tan, Jiazi, Lixia Yang, Alan Ann Lerk Ong, Jiahao Shi, Zhensheng Zhong, Mun Leng Lye, Shiyi Liu, et al. 2019. “Wrong- A Disease-Causing Intronic Point Mutation C19G Alters Tau Exon 10 Splicing via RNA Secondary Structure Rearrangement - Wrong Listing.” *Biochemistry*. https://doi.org/10.1021/acs.biochem.9b00001.

Woods, Chanin T., Lela Lackey, Benfeard Williams, Nikolay V. Dokholyan, David Gotz, and Alain Laederach. 2017. “Comparative Visualization of the RNA Suboptimal Conformational Ensemble In Vivo.” *Biophysical Journal* 113 (2): 290–301. https://doi.org/10.1016/j.bpj.2017.05.031.